# Paradoxical noise preference in RNNs

**Noah Eckstein**                                                    *eckstein.81@osu.edu*
*Department of Mechanical and Aerospace Engineering*
*The Ohio State University*

**Manoj Srinivasan**                                                 *srinivasan.88@osu.edu*
*Department of Mechanical and Aerospace Engineering*
*Program in Biophysics*
*The Ohio State University*

**Reviewed on OpenReview:** *https://openreview.net/forum?id=gqxTZRzI35*

## Abstract

In recurrent neural networks (RNNs) used to model biological neural networks, noise is typically introduced during training to emulate biological variability and regularize learning. The expectation is that removing the noise at test time should preserve or improve performance. Contrary to this intuition, we find that continuous-time recurrent neural networks (CTRNNs) often perform best at a nonzero noise level, often approximately the same level used during training. This noise preference typically arises when noise is injected inside the neural activation function; networks trained with noise injected outside the activation function perform best with zero noise. The phenomenon arises robustly in diverse tasks for large enough training noise including function approximation, maze navigation, 2D path integration, and a multi-task suite from cognitive neuroscience; we also show the phenomenon arising in feedforward neural networks, not just in RNNs. Through analyses of simple function-approximation and single-neuron regulator tasks, we show that the phenomenon stems from noise-induced shifts of fixed points (stationary distributions) in the underlying stochastic dynamics of the RNNs, thereby providing some mechanistic interpretability of the phenomenon. These fixed point shifts are noise-level dependent and bias the network outputs when the noise is removed, degrading performance. Analytical and numerical results show that the bias arises when neural states operate near activation-function nonlinearities, where noise is asymmetrically attenuated, and that performance optimization incentivizes operation near these nonlinearities; such performance incentives exist for networks with noise inside the activation function, but not for networks with noise outside the activation function, explaining why only noise-in networks show preference. Thus, networks can overfit to the stochastic training environment itself rather than just to the input–output data. The phenomenon is distinct from stochastic resonance, wherein nonzero noise enhances signal processing. Our findings reveal that training noise can become an integral part of the computation learned by recurrent networks, with implications for understanding neural population dynamics and for the design of robust artificial RNNs.

## 1 Introduction

Recurrent neural networks (RNNs) are central in computational neuroscience as models of population-level computation (Güçlü & Van Gerven, 2017; Yang & Molano-Mazón, 2021; Barak, 2017; Driscoll et al., 2024; Yang et al., 2019; Wang et al., 2021). Noise is commonly introduced during the training of these RNNs (e.g., Yang et al. (2019); Driscoll et al. (2024)). One major reason for this is to model the noisy environment in which biological neurons must operate (Faisal et al., 2008), as multiple notable neural phenomena have been shown to result from mechanisms that only emerge to reject or otherwise address noise (e.g., Burak & Fiete (2012); Krishna et al. (2024)). Similarly, noise is essential for modeling the generative behavior of

biological neural networks (Bredenberg et al., 2026). Another major reason to add noise during training is for regularization (Lim et al., 2021). Such noise promotes stable training and reduces sensitivity to individual neural activities or weights, discouraging sharp minima and effectively acting as a complexity penalty (Bishop, 1995; You et al., 2019; Noh et al., 2017; Lim et al., 2021). If noise served only as regularization, removing it at test time should maintain or improve performance. Contrary to this expectation, we find that continuous-time recurrent neural networks (CTRNNs; Yang et al. (2019); Driscoll et al. (2024), Figure 1) perform best at a nonzero noise level, typically matching the training noise. Thus, the networks develop a preference for a particular noise level. We analyze the dynamical basis of this noise preference, showing it arises from how noise shifts the stationary distributions of the underlying stochastic dynamics, suggesting a mechanistic basis of the phenomenon. We examine different noise injection schemes to identify conditions under which a preference naturally develops. Though reminiscent of stochastic resonance-like phenomena in which noise enhances signal processing (Katada & Nishimura, 2009; Metzner et al., 2024; Krauss et al., 2019), our noise preference phenomenon is mechanistically distinct, implying that noise-facilitated behavior may have a broader computational role in neural systems.

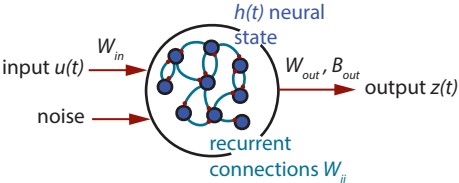

Figure 1: A continuous-time recurrent neural network with input and output at each time. Variants of this network are used in computational neuroscience (Yang et al., 2019; Driscoll et al., 2024), and the effect of synaptic noise on such networks is studied here.

## 2 Methods

### 2.1 CTRNN architecture

CTRNNs (Figure 1) implement ordinary differential equations that coarsely model synaptic interactions among a population of biological neurons (Barak, 2017; Yang et al., 2019; Driscoll et al., 2024). Their internal dynamics are usually described by the equation:

$$\tau\frac{dh}{dt} = -h + f(W_{\text{rec}}h + W_{\text{in}}u + B_{\text{in}} + \sigma_{\text{in}}\eta), \tag{1}$$

where $h \in \mathbb{R}^n$ is a vector denoting the neural state (i.e., the state of the network), usually modeling neuronal firing rates in biological networks (Yang et al., 2019), $n$ is the number of neurons, $\tau \in \mathbb{R}$ is the neural time constant, $W_{\text{rec}} \in \mathbb{R}^{n \times n}$ is the recurrent weight matrix, $u \in \mathbb{R}^m$ is the vector containing various inputs (e.g., sensory feedback, task identity), $W_{\text{in}} \in \mathbb{R}^{n \times m}$ is the input weight matrix, $B_{\text{in}} \in \mathbb{R}^n$ is the neural bias vector, $\sigma_{\text{in}}\eta \in \mathbb{R}^n$ is a Gaussian white noise process with variance $\sigma_{\text{in}}^2$ along all dimensions, and $f(\cdot)$ is a scalar nonlinear activation function, applied element-wise over vector arguments. The networks produce outputs $z \in \mathbb{R}^p$ via an affine transform $z = W_{\text{out}}h + B_{\text{out}}$, where $W_{\text{out}} \in \mathbb{R}^{p \times n}$ and $B_{\text{out}} \in \mathbb{R}^p$ are the output weight matrix and bias vector, respectively.

To simulate these networks, we integrate Equation 1 by Euler's method (as in Yang et al. (2019) and others), giving the discrete time equation:

$$h_{t+1} = (1 - \gamma)h_t + \gamma f(W_{\text{rec}}h_t + W_{\text{in}}u_t + B_{\text{in}} + \mathcal{N}(0, \sigma_{\text{in}}^2)), \tag{2}$$

where $\gamma = \Delta t/\tau$ is a non-dimensional ratio of $\Delta t$, the discrete timestep, and $\tau$, the neural time constant, and $\sigma_{\text{in}}$ is the standard deviation of the injected noise, which is sampled independently for each neuron in each timestep. Unless otherwise stated, we used $\tau = 0.1$ and $\Delta t = 0.02$, resulting in $\gamma = 0.2$. For $f(\cdot)$, most

of our experiments use rectifying activation functions, either ReLU or its smooth approximation, SoftPlus (see Section 2.2.5 for one exception).

As presented thus far, the noise is injected inside the activation function, as is common in the computational neuroscience literature (e.g., Yang et al. (2019); Driscoll et al. (2024)). This placement ensures positivity of the neural state $h$, facilitating their interpretation as firing rates. However, some important modeling work in computational neuroscience uses noise injected outside the activation function (Burak & Fiete, 2012; Bredenberg et al., 2026; Krishna et al., 2024), as such formulations may still permit rate-based interpretation as long as neural activations are large in comparison to the noise. Thus, we also consider a variant of Equation 2 with the noise added outside the activation function:

$$h_{t+1} = (1 - \gamma)h_t + \gamma(f(W_{\text{rec}}h_t + W_{\text{in}}u_t + B_{\text{in}}) + \mathcal{N}(0, \sigma_{\text{out}}^2)). \tag{3}$$

We refer to networks trained with noise injected inside and outside their activation functions as noise-in and noise-out networks, respectively, and we refer to their noise as either pre-activation noise (for noise-in networks) or post-activation noise (for noise-out networks).

## 2.2 Tasks for illustrating the effect of noise

To study the effect of noise in CTRNNs (Figure 1), we trained both noise-in and noise-out variants on simple tasks and evaluated their performance, focusing on whether or not there is a nonzero noise level for optimal performance at test time. We mainly focus on a simple function approximation task to illustrate the phenomenon, but we also include a single-neuron regulator task to build intuition and three more complex tasks/task families used in cognitive neuroscience to test whether the results we see in these simpler tasks are general enough to apply in more neuroscientifically relevant contexts. All these tasks are supervised learning tasks, trained by minimizing a mean squared error loss function; see Appendix A for further training details. We provide the essential elements of the tasks in the remainder of this section; see Appendix B for detailed equations and implementation information.

### 2.2.1 Function computation tasks

In the function computation tasks, we require the neural networks to compute univariate functions ($g(x)$, $x \in \mathbb{R}$, $g(x) \in \mathbb{R}$) over some compact interval. The networks receive a constant one-dimensional input $x$ for the argument of the function being approximated, and the network output at the end of a "computation" period must match the corresponding value of the function $g(x)$. We tested with two slightly different readout procedures: one with a fixed readout time and another where the readout time was uniformly random within a final interval, forcing the network to maintain the correct output $g(x)$ across multiple steps. We considered two different target functions: $g(x) = \sin(x)$ for $x \in [0, 2\pi]$ and $g(x) = \tanh(x)$ for $x \in [-4, 4]$.

### 2.2.2 Maze navigation task

Maze navigation tasks have a long history in behavioral and systems neuroscience (Olton, 1979; Ainge et al., 2007). In our maze navigation task, the recurrent neural network acts as an instantaneous velocity controller for a particle in a fixed maze environment (Figure 3a). The objective is to navigate the particle from one maze vertex (the blue X's in Figure 3a), to another, pausing at the vertices along the way. The particle is perturbed by random velocity fluctuations that the network must reject as it navigates. In each trial, the starting and destination particle positions are sampled uniformly at random from the list of maze vertices. The networks receive constant inputs communicating the coordinates of their destination vertex, the current position of the particle, and an indication of whether the network should be paused at a vertex or traveling from one vertex to the next.

### 2.2.3 Multi-cognitive task suite

RNNs trained to perform many different tasks in different contexts have been of interest to the computational neuroscience community as a kind of "model organism" for studying mechanisms for flexible cognitive control (Yang et al., 2019; Driscoll et al., 2024; Khona et al., 2023). In this work, we trained RNNs on a subset of the

20 tasks popularized by Yang et al. (2019), all of which require networks to manipulate 2D directional stimuli and/or responses in imitation of canonical animal cognitive tasks with saccadic eye movements as the trained response modality. Of the original set of 20 tasks, we chose the following 6 (here, we use the task names from Driscoll et al. (2024)). **ReactPro/ReactAnti:** In the pro version, networks must produce an output that matches the directional stimulus as soon as the stimulus appears. In the anti version, the network output must be opposite the presented stimulus. **MemoryPro/MemoryAnti:** These are similar to the React tasks, but the outputs must be produced after a "memory period" of variable length, during which the stimulus is removed, and the network must wait for a cue before releasing its output; **DelayPro/DelayAnti:** These are similar to the Memory tasks, but the stimulus remains on, and the network must release its output with a fixed delay after stimulus onset and without any cueing. We follow the training procedure and task implementation of Driscoll et al. (2024), with some slight deviations in the training procedure and task design as described in Appendices A and B.

### 2.2.4 Path integration task

Path integration tasks, that is, tasks in which organisms must maintain an accurate estimate of their spatial position and/or orientation by integrating self-motion cues, have been an influential source of evidence in the study of spatial cognition and its neural substrates, both in experimental and computational neuroscience (Samsonovich & McNaughton, 1997; McNaughton et al., 2006; Cueva & Wei, 2018; Sorscher et al., 2023). In this work, we trained RNNs on a version of the path integration task studied in Cueva & Wei (2018). The network observes a particle that is randomly traversing a two-dimensional square room, receiving sensory inputs for speed and heading at every timestep, and it must output estimates of the current position of the particle within the room by integrating this sensory information. See Appendices A and B for training and task details.

### 2.2.5 Single-neuron regulator task

To illustrate the effect of noise in a simpler task with fewer parameters so we could more easily isolate mechanisms, we consider a simple regulator task (Anderson & Moore, 2007), so simple that a network with only one neuron recurrently connected to itself, no inputs, and direct neural output (that is, the output is just the neural state $h_t$) can do it. In this single-neuron regulator task, the neural state must match a constant setpoint $r$ in the presence of noise, which, with some added complexities, is also the essential objective of the function computation, maze navigation, and multi-cognitive tasks. For our experiments with the single-neuron regulator task, we injected both pre- and post-activation noise, subsuming Equations 2 and 3 as special cases; we varied the proportion of these noise levels to observe how the loss function landscape varies as the noise goes from primarily pre-activation to primarily post-activation. As we will show, the noise type (e.g., pre- or post-activation) is a key independent variable in determining the final role of the noise in the trained networks. With only one neuron and no inputs or explicitly computed outputs, the only trainable parameters are the scalar recurrent weight and the scalar neural bias. The task itself is effectively parameterized by only two dimensionless parameters: the ratio of the pre-activation noise standard deviation to the post-activation noise standard deviation ($\sigma_{\text{in}}/\sigma_{\text{out}}$) and the ratio of the setpoint value to the post-activation noise standard deviation ($r/\sigma_{\text{out}}$). For these experiments, we tried both ReLU and tanh activation functions to assess the role of rectification.

## 3 Results

### 3.1 Function computation networks trained with noise inside, not outside, their activation functions require noise for best performance.

We trained both noise-in and noise-out networks (Equations 2 and 3) on the function computation tasks and tested them under a range of noise levels, including zero noise at the low end. For noise-in networks trained with sufficiently high training noise (Figure 2b), root mean squared error (RMSE) was minimized when the test noise roughly matched the training noise (Figure 2a). This shows that the noise does more than act as a regularizer: it becomes integral to the computation. This noise-preference phenomenon emerges

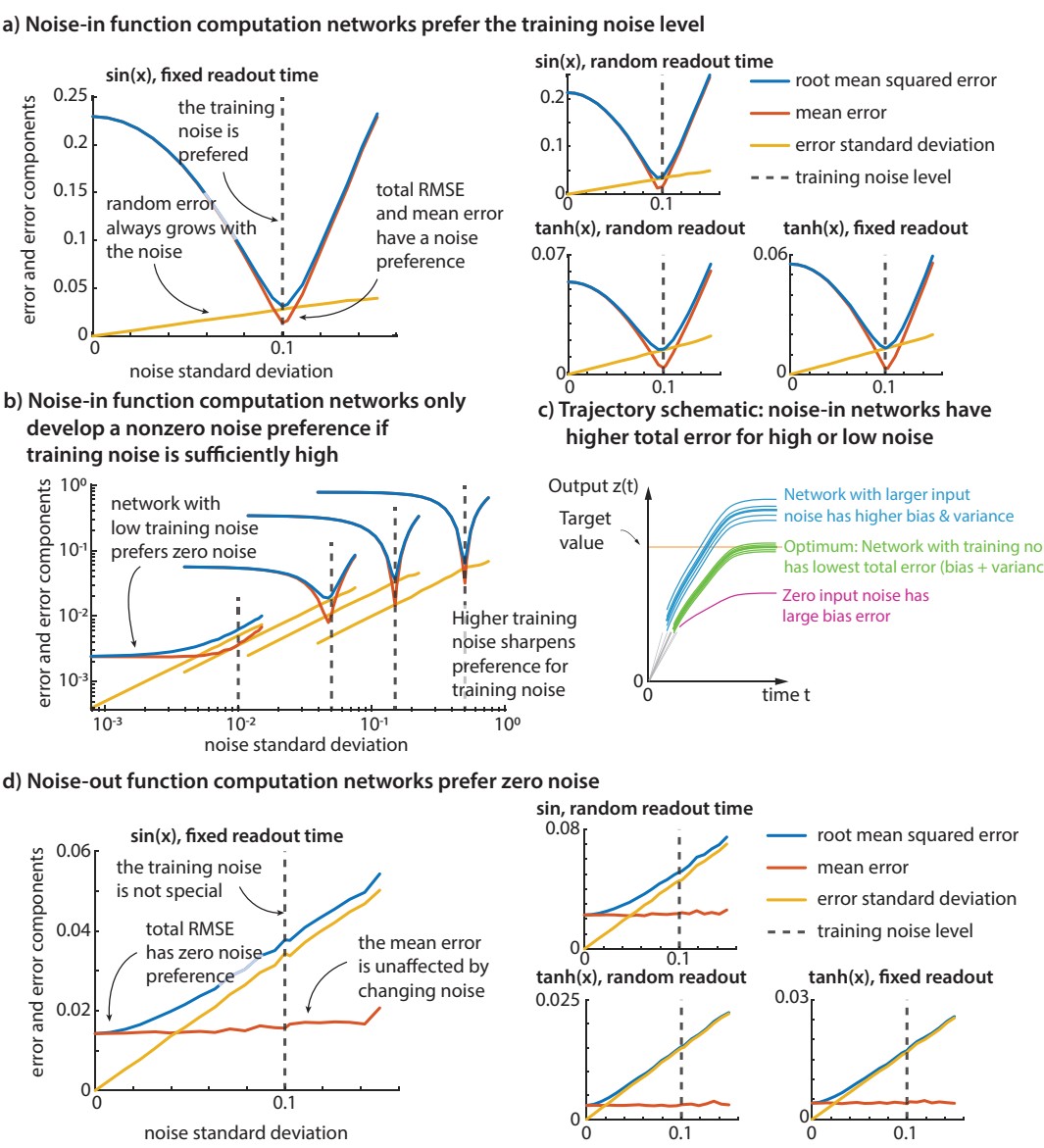

Figure 2: **Simple function computation: Noise-in networks prefer the training noise level, but noise-out networks prefer zero noise.** a) Root mean squared error, mean error, and error standard deviations for noise-in function computation networks. The best performance occurs when testing at or near the training noise level, and this preference is driven by a non-monotonic dependence between the noise level and the mean error (i.e., the systematic error). This indicates that passing the noise through the activation function somehow induces a noise-level-dependent systematic bias in the network outputs. b) Noise-preference results (same color meanings as in previous panel) for fixed readout time sine computation networks trained with various training noise levels. Noise preference becomes more pronounced for greater training noise, with very low training noise resulting in zero noise preference. Overall performance at the preferred noise gets worse as training noise increases. Note the logarithmic axes. c) Schematic illustrating that the network output distribution has the lowest total squared error at the training noise despite a larger error variance than the zero noise case. The mean trajectory is depicted by a thick line and individual realizations by thin lines. d) Root mean squared error, mean error, and error standard deviations for noise-out function computation networks. Unlike the noise-in networks, the noise-out networks perform best when tested with zero noise, and the systematic error appears unaffected by changes in the noise.

repeatably, for instance, networks trained independently with different initializations have similar noise-preference curves (Appendix Figure A1). In contrast, noise-out networks performed best with zero noise (Figure 2d and Appendix Figure A2), consistent with the noise serving a purely regularizing role. In noise-in networks (Figure 2a-b), the increase in performance deficits away from the training noise levels were largely driven by bias rather than variance. While error variance grew roughly linearly with the variance of the injected noise, systematic error (bias) changed non-monotonically, being high for both zero noise and for noise much higher than training noise. The non-monotonicity in the bias error is qualitatively similar to the non-monotonicity in the total error, except that the bias error is minimized at a slightly higher noise level than the total error. This suggests that noise-dependent output biases, linked to the activation function, cause the observed preference.

## 3.2 Noise preference also emerges in more complex, neuroscientifically relevant networks and simple feedforward networks.

The maze navigation task, multi-cognitive task suite, and path integration task confirmed that this effect generalizes beyond simple function approximation. Noise-in networks trained on these three task families also achieved their best performance when tested at a nonzero noise level close to the training noise level (Figure 3b-d), although, in detail, the optimal performance for the multi-cognitive task suite and the path integration was further away from and generally smaller than the training noise level. Just as for the function computation RNNs, significant deviations from the training noise produced performance deficits, and this non-monotonicity of total error is largely driven by error bias, not error variance. Specifically, in these RNN experiments, we find that the error bias is optimal at nearly the training noise level and the error standard deviation is monotonic increasing: these two observations imply that the total error will have a minimum to the left of the training noise level as seen in these experiments. These more complex tasks show that the noise preference mechanism persists in more realistic, neuroscientifically rich settings. We also trained simple noise-in feedforward multilayer perceptrons (MLPs) on a function computation task (only the sine function, see Appendix A for network and training details) to see if the phenomenon is specific to recurrent networks or if it might emerge in a network with no explicit temporal dynamics; we found that a noise preference, again driven by error bias, emerges here as well (Figure 3d). Unlike in the RNNs, for these feedforward networks, the error bias was minimized at slightly higher than the training noise level. Taken together, these results suggest the noise preference phenomenon could be of interest for multiple areas within the broader landscape of neural network research.

## 3.3 Fixed points of stochastic RNNs can be shifted by noise

Prior work has shown that CTRNNs solve tasks through low-dimensional dynamical features such as fixed points (Driscoll et al., 2024; Yang et al., 2019; Vyas et al., 2020). For a deterministic dynamical system with constant inputs, fixed points are invariant points in state space with $\dot{h} = 0$; that is, a network starting at a fixed point remains there forever in the absence of perturbations (e.g., point attractors are asymptotically stable fixed points). Both our function computation and maze tasks require maintaining static outputs over time, making asymptotically stable fixed points natural computational primitives. While there are no true fixed points for the stochastic neural networks (Equations 1-3), for these systems, we use "fixed point" to refer to the mean of the stationary distribution near point attractors.

In the following subsections, we first demonstrate how both pre- and post-activation noise can induce shifts in the effective fixed point locations of a noisy CTRNN. Then, in Section 3.4, we demonstrate that these shifts quantitatively predict noise preference behavior, thus providing a dynamical mechanism for the phenomenon.

### 3.3.1 Noise can shift fixed points in noise-in networks

In noise-in networks, the activation function $f(\cdot)$ is applied after the noise samples are added to the summed and biased synaptic inputs, namely, $W_{\text{rec}}h_t + W_{\text{in}}u_t + B_{\text{in}}$ (Equation 2). Therefore, for a single neuron in a noise-in network that has settled into a stationary state distribution, the argument to the activation function will be a sum of samples from two distributions, one corresponding to the summed and biased synaptic input, which generally will come from a distribution with nonzero mean $\mu_s$, and one corresponding to the injected

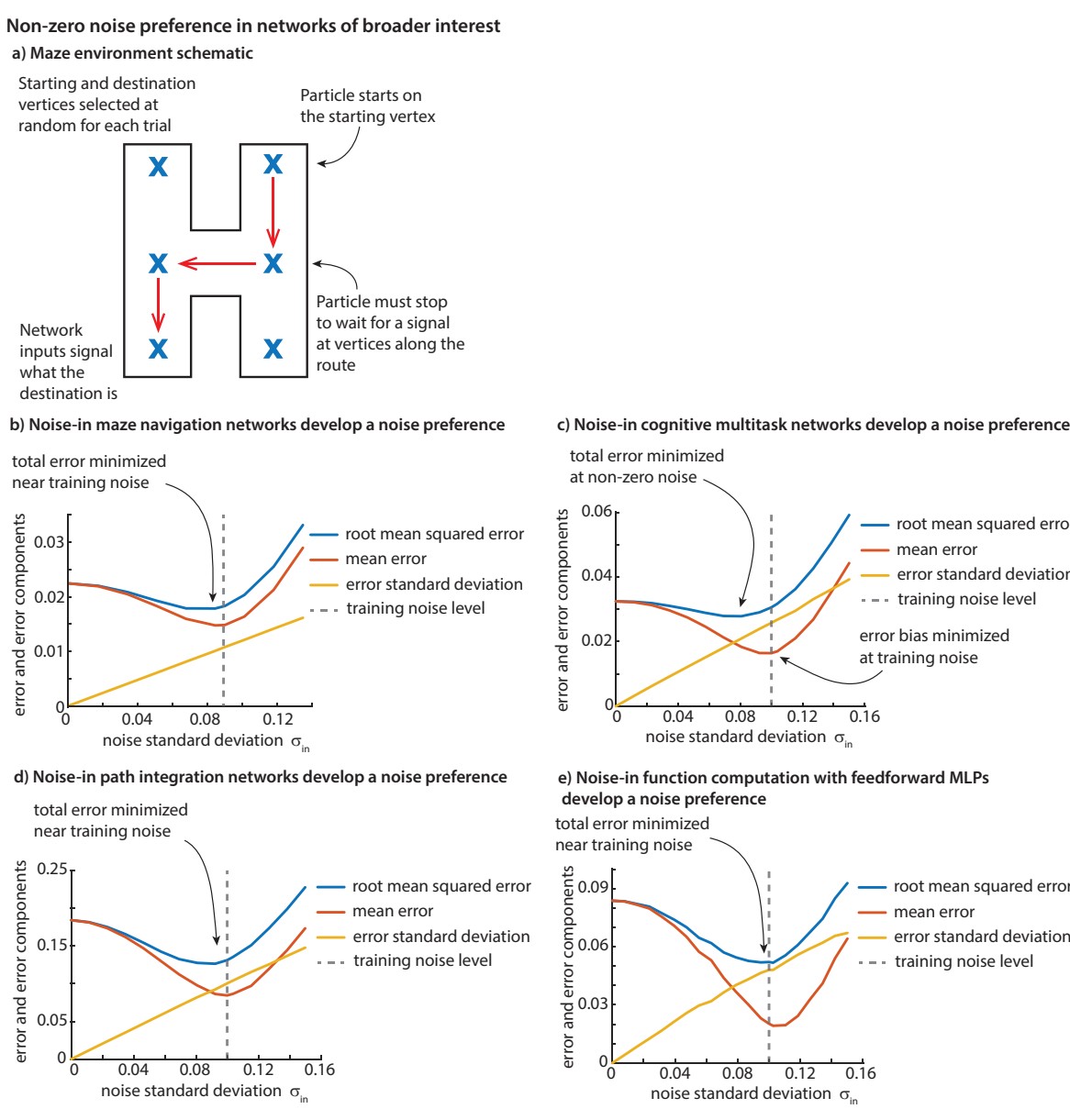

Figure 3: **Maze navigation task schematic and noise preference results for assorted cases of broader interest.** a) Schematic of the maze environment. The blue x's show the locations of maze vertices where the network has to stop and wait as it navigates from the start vertex to the destination vertex. The red arrows between vertices denote the multiple short jaunts that make up one of the routes through the maze. b) Root mean squared error, mean error, and error standard deviation for a noise-in maze navigation network when tested at a variety of different noise levels. As with the function computation networks, the network performs best near the training noise level, and the performance deficits at other noise levels are driven by error bias, not variance. c) Just as in the previous networks, noise-in networks trained on the multi-cognitive task suite develop a noise preference. Here, the optimal performance occurs at a noise level more distant from the training noise, but this difference is due to greater error variance at higher noise levels; the error bias still clearly favors a noise level close to the training noise. d) Noise-in path integration networks also develop a noise preference, demonstrating that the phenomenon occurs in networks that use higher-dimensional attractors, not just point attractors. e) Even feedforward multilayer perceptrons trained on the function computation task exhibit a noise preference, demonstrating that this phenomenon is relevant outside the context of time-evolving networks. Error statistics are from an ensemble of simulations with different noise realizations; in particular, the error standard deviation indicates this trial-to-trial variance.

noise, which by design will come from a distribution with zero mean $\mathcal{N}(0, \sigma_{\text{in}}^2)$. Ignoring the synaptic input variability, the excitatory term in the discrete update rule for this neuron is then proportional to $f(\mu_s + \mathcal{N}(0, \sigma_{\text{in}}^2))$. (If we do not ignore variability in the synaptic input term, we can just lump that variability with the injected noise term to reach the same qualitative conclusions.) If $\mu_s$ is large in comparison to $\sigma_{\text{in}}$ and negative, the noise will only very rarely push the argument of the activation function high enough to excite the neuron, effectively making it such that the noise does not affect this neuron. Similarly, if $\mu_s$ is large and positive, the noise will only very rarely push the argument of the activation function low enough to significantly involve the saturation of the activation function; this makes the noise act as excitation and inhibition in equal proportion, resulting in zero net effect on the expected state of the neuron. If $|\mu_s|$ is on the order of $\sigma_{\text{in}}$, however, the argument regularly fluctuates in and out of the saturated region, resulting in the selective attenuation of inhibitory noise samples. In this situation, the noise has a net excitatory effect on the expected activation level of the neuron, and this effect is $\sigma_{\text{in}}$-dependent (Figure 4a), because a larger $\sigma_{\text{in}}$ results in more frequent, larger magnitude crossings into and out of the saturated region.

Expanding this reasoning to all the neurons in the network, we see that the locations of fixed points situated near boundaries between linear regions of the dynamics (where some neurons receive a net excitatory drive from the noise) are noise-level dependent. If the network has been fully optimized for some task that depends on the location of such a fixed point, we might expect noise level changes at test time to worsen performance.

### 3.3.2 Noise can shift fixed points in noise-out networks

Consider the piecewise-linear differential equation given by:

$$\dot{h} = \begin{cases} -\theta_{large}h + \sigma\eta & \text{when } x \leq 0 \text{ and} \\ -\theta_{small}h + \sigma\eta & \text{when } x > 0 \end{cases} \tag{4}$$

where $0 < \theta_{small} < \theta_{large}$, and $\eta$ is a white noise process. This is a piecewise version of an Ornstein-Uhlenbeck (OU) process (Uhlenbeck & Ornstein, 1930), and we can implement it in a noise-out, one-neuron CTRNN with ReLU activation and no inputs or activation bias as follows:

$$\tau\frac{dh}{dt} = -h + \text{ReLU}(w_{\text{rec}}h) + \frac{\sigma}{\theta_{small}}\eta, \text{ with } \tau = \frac{1}{\theta_{small}}, \text{ and } w_{\text{rec}} = -\frac{\theta_{large} - \theta_{small}}{\theta_{small}} \tag{5}$$

In the absence of noise ($\sigma_{\text{out}} = 0$), this system has a single stable fixed point at $h = 0$ (i.e., its fixed point is exactly on the boundary between linear regions), but for $\sigma_{\text{out}} > 0$, the stationary distribution acquires an increasing positive shift as the noise level increases (Figure 4b). This is because noise-induced deviations in the positive direction persist for longer owing to the smaller decay rate on that side, and the larger the deviations are, the greater the discrepancy between the positive and negative decay times tends to be.

Similar to the situation for noise-in networks, this emergent shift in the stationary distribution depends on the fixed point being close to or coincident with the boundary between linear regions. If we add a large negative neural bias term inside the activation function in Equation 5, it would push the piecewise boundary to some large negative $h$ value while leaving the fixed point at $h = 0$. In this situation, there would be a negligible noise-induced shift in the stationary distribution because the dynamics would become equivalent to the standard Ornstein-Uhlenbeck process except in the rare case of very large negative excursions from the fixed point. We would get a similar result if we included a large positive neural bias, but in that case, both the piecewise boundary and the fixed point would move in the positive direction, but the boundary would move farther, producing a gap between the two.

### 3.4 Noise-induced fixed point shifts predict low-noise performance deficits

To investigate the possibility that fixed point shifts are responsible for the noise preference, we first computed the fixed points of our noise-in function computation networks, both with and without accounting for noise bias effects; see Appendix C for the fixed point finding procedure. We computed the difference between the zero noise and training noise fixed point locations, and we projected these training-noise-induced fixed point shifts by the transformation that produces the output ($z = W_{\text{out}}h + B_{\text{out}}$) to compute the resulting shifts in

**Noise can effectively shift the fixed points in both noise-in and noise-out networks**

**a) Noise-in neuron: Mean excitation increases with the noise level when near saturation**

**b) Noise-out network: Noise shifts the effective fixed point when a fixed point is between linear regions.**

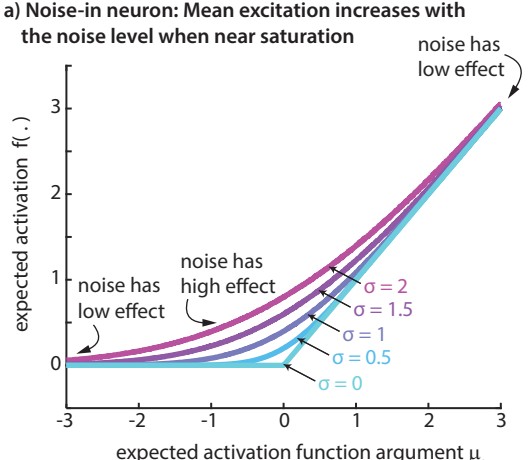

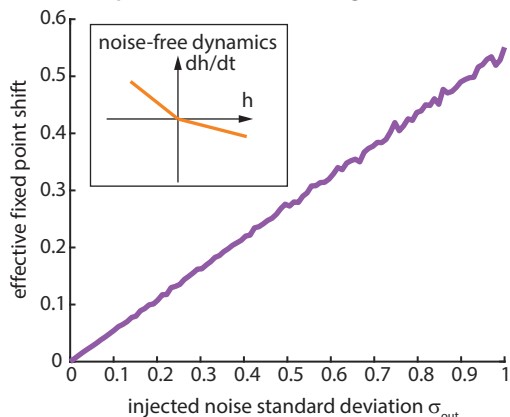

Figure 4: **Noise can shift the fixed points of both noise-in and noise-out networks if they have fixed points near a boundary between linear regions of their dynamics.** a) Functional relationship between the mean of a normal distribution before and after transformation by the ReLU activation function for multiple different standard deviations. When the absolute value of the mean is on the order of the standard deviation, interaction with the nonlinearity causes a variance-dependent positive shift in the mean of the transformed distribution. As a result, the locations of fixed points in a noise-in network will depend on the noise level if those fixed points are situated near boundaries between linear regions in the dynamics. b) Functional relationship between the mean of the stationary distribution of a piecewise Ornstein-Uhlenbeck process (Equations 4 and 5) and the standard deviation of the noise. Inset shows a noise-free single-neuron version of such a noise-out network. As the standard deviation grows, the stationary distribution shifts in the positive direction because deviations in that direction take longer to decay. This single-neuron example illustrates why the locations of fixed points in a noise-out network depend on the noise level if those fixed points are situated near boundaries between linear regions.

the output. We found that these output shifts approximately predict the input-specific error biases observed when testing the networks with zero noise (Figure 5).

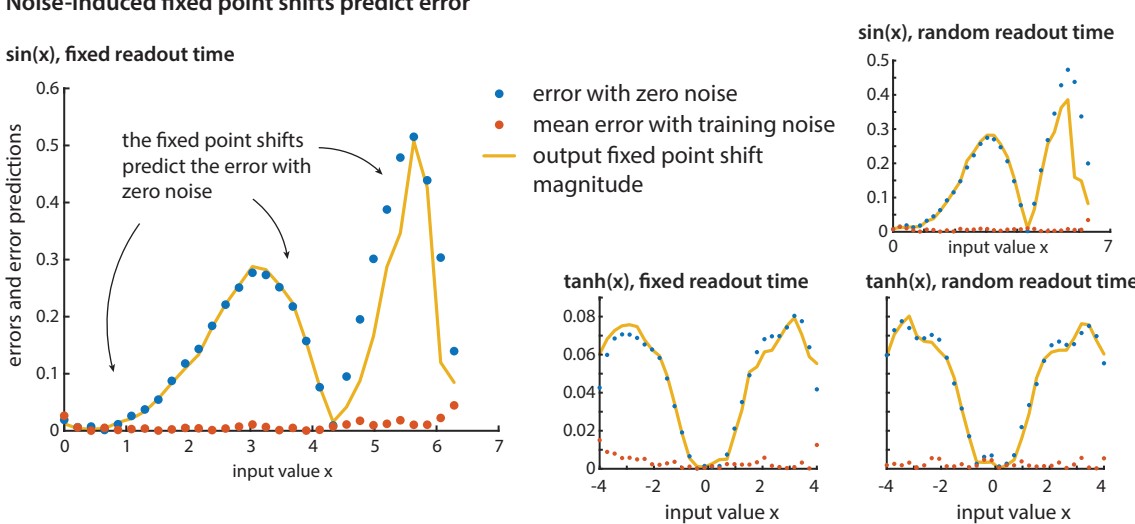

Figure 5: **Noise-induced fixed point shifts predict input-specific error in noise-in function computation networks tested with zero noise.** Noise induced fixed point shifts projected onto the output matrix closely track the zero noise error profile across the input domain, confirming that noise-induced fixed point shifts are responsible for the noise preference.

### 3.5 Single-neuron regulator: noise inside the activation function alters the loss landscape to favor solutions that rely on noise-induced fixed point shifts

Since both noise-in and noise-out networks can exhibit noise-dependent fixed point shifts, but we only see a noise preference in noise-in networks, we hypothesized that there may be some performance incentive specific to noise-in networks that drives the development of a preference. Given our reasoning in Sections 3.3.1 and 3.3.2, we sought to understand how the loss function varies with the distance between fixed points and nonlinear boundaries in the dynamics. For simplicity, we use the single-neuron regulator task to investigate this.

In the single-neuron regulator task, the setpoint $r$ determines where the neuron will attempt to center its stationary distribution. So by varying the setpoint, we control how close or how far the stationary distribution is from the saturation of the activation function. For the ReLU activation function, large positive setpoints encourage the stationary distribution to be far from the saturation (where we do not expect noise preference), while small positive setpoints encourage stationary distributions near the boundary of the saturation (where we may expect a noise preference). For the tanh activation function, setpoints near zero encourage stationary distributions far from either saturation boundary, while setpoints with absolute value close to one encourage stationary distributions near one of the saturation boundaries. We trained many single-neuron regulators with either the ReLU or tanh activation functions, each with a different combination of setpoint and pre-activation noise level, setting the post-activation noise level to one without loss of generality (implicitly scaling the problem by this noise level). This allowed us to observe which setpoints permit the best performance given the relative strength of the pre- and post-activation noise streams. We optimize the bias for various fixed values of the recurrent weight because training both the recurrent weight and the neural bias resulted in highly discrete, overshooting dynamics, which we do not observe in the other, more complicated networks.

The single-neuron regulator experiments showed that there are regions in the parameter space where networks (both ReLU and tanh) exhibit a noise preference. When the pre-activation noise dominates the post-activation noise, the best networks tend to land inside or near these regions with a noise preference, but when the post-activation noise dominates, they land squarely outside these regions (Figure 6). We suggest

that this trend is due to a tradeoff. If the stationary distribution straddles the saturation boundary, a portion of the pre-activation noise gets turned off, while leaving a portion of the post-activation noise out of reach for the recurrent weight to help in drawing the state back toward its expected value. Therefore, by placing the stationary distribution near saturation points, the network can reduce the performance degradation caused by pre-activation noise at the price of exacerbating the performance degradation caused by post-activation noise (note that in our single neuron case, the price of operation near saturation is just the increased influence of post-activation noise, but for multi-neuron networks without post-activation noise like the various other networks we present here, the price is really a loss of coupling between neurons, as saturated neurons become insensitive to their synaptic inputs). For a network dominated by pre-activation noise, this is a good trade, so the stationary distribution should be placed near saturation, but for a network dominated by post-activation noise, this is a bad trade, so the stationary distribution should be placed far from saturation. This is consistent with the observation that networks with higher magnitude recurrent weights tolerate higher inside noise levels before opting to place their setpoints near saturation boundaries (Figure 6), as networks with a larger recurrent weight are better able to reject noise without relying on the noise attenuation capability of the activation function; therefore, the pre-activation noise needs to be significantly larger than the post-activation noise to incentivize the network to forgo the noise attenuation afforded by the recurrent weight. Given the reasoning described in Sections 3.3.1-3.3.2 and illustrated in Figure 4, which establishes that operation near saturation boundaries induces biases in network latents, and the evidence presented in Section 3.4 and Figure 5, which establishes that the noise preference in our function computation networks was likely a downstream result of such biases, this tradeoff we describe is the likely cause for the discrepancy we observed between noise-in networks and noise-out networks. Put simply, noise-out networks tend to perform better on tasks involving output variance reduction if they operate neurons far from saturation (where we *do not* expect a noise preference), while noise-in networks tend to perform better on such tasks if they operate neurons near saturation (where we *do* expect a noise preference), and this is because noise-in networks can leverage the activation function saturation for noise attenuation, while noise-out networks cannot. Further, this reasoning does not require that the activation function be rectifying, simply that it should be saturating, which is consistent with the presence of the phenomenon in both ReLU and tanh single-neuron regulator experiments (Figure 6a-b).

## 4 Discussion

Using a simple function computation framework, a maze navigation framework, a standard multitask framework from cognitive neuroscience, and a path integration framework as illustrative examples, we have shown that noisy CTRNNs sometimes develop a preference for a specific optimal noise level, equal to or close to that used during training: changes in the noise level at test time result in worse performance. When noise is injected before the application of the activation function, the networks tend to develop a noise preference, but when noise is injected after, they do not. By analyzing the fixed points of the noise-in function computation networks, we found that the noise preference is driven by noise-variance-dependent shifts in the neural state distributions. These shifts are caused by inhomogeneous noise attenuation that only occurs when the neural state is near a boundary between linear regions of the network dynamics. Though the locations of the stationary distributions of both noise-in and noise-out networks will depend on the noise if they are placed near such boundaries, experiments with a single-neuron regulator problem showed that only the noise-in networks have a performance incentive to do so, thus explaining why only the noise-in networks developed a noise preference.

Our findings demonstrate that training noise can become an integral component of the computation learned by a network, shaping dynamical structure and inducing systematic output biases that render performance intrinsically dependent on the training noise level rather than merely regularized by it. For computational neuroscience, this suggests a concrete hypothesis: neural circuits operating near nonlinear activation boundaries may leverage—or become functionally dependent on—synaptic variability to stabilize activity and reduce variance, implying that some observed noise sensitivity in biological systems could reflect learned dynamical structure rather than incidental variability. For machine learning researchers, the results highlight a subtle failure mode of noise injection, namely overfitting to the stochastic training environment itself, and thereby contribute to the broader scientific objective of understanding how dynamical systems perform

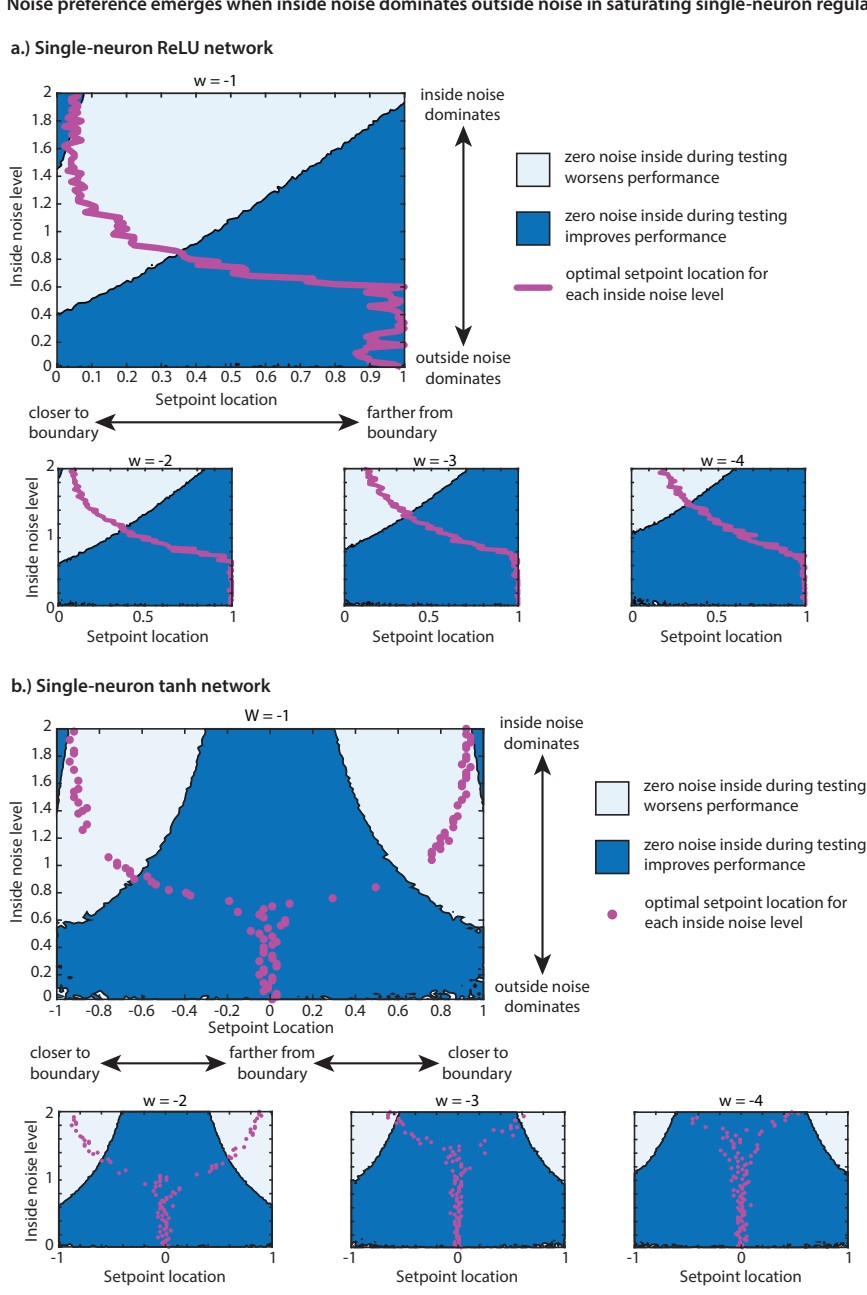

Figure 6: **Single neuron regulator experiments show that increasing inside noise-level encourages setpoints closer to the saturation boundaries** a) Using ReLU activation function. As the pre-activation noise grows in relation to the post-activation noise, the optimal setpoint gets closer to the saturation boundary ($r = 0$ for ReLU, $r = \pm 1$ for tanh), eventually getting close enough to produce a noise preference. This is true for all values of the recurrent weight, $w$, that we sampled except for $w = -4$ in the tanh network (though slightly higher inside noise would likely produce a noise preference). This explains why we see a noise preference in the noise-in (and not noise-out) function computation networks. b) Using tanh activation function also results in noise preference: because tanh saturates for both high and low input values, we correspondingly find noise preference when the setpoint location is high or low.

computation under noise, clarifying when variability serves as a stabilizer or regularizer versus a hidden dependency.

## 4.1 Noise as regularization

Our analyses focused on the role noise might play in computation beyond regularization, and we found that indeed, for noise-out networks, the noise can become an essential part of the computation (Figure 2a), with greater training noise levels resulting in sharper noise preference (Figure 2b). Given our explanation for this noise preference phenomenon, which hinges on the networks being incentivized to reduce activation variance caused by the noise, this trend, where greater training noise induces a sharper noise preference, makes sense. Less noise means networks have less to gain by operating neurons near saturation. However, we also found that performance at the preferred noise level becomes worse as the training noise is increased for the simple function computation networks we studied (Figure 2b). This suggests that, in these simple networks, the noise does not actually serve as a regularizer, or at least, it does not confer the test performance advantage we would hope of a regularizer. We point out that regularization is not expected to improve performance on the training set. On the contrary, regularization is expected to improve performance on the test set by reducing model conformity to the tails of the training distribution, which necessarily means decreasing training performance. For these function computation tasks, and for all tasks explored in this work, there was no notion of a test set; during both training and testing, batches were sampled generatively from the same task distribution (which, in all tasks, lacked particularly heavy tails), so the training performance deficit that one expects to result from regularization applies to testing as well. In such cases, the best we can hope for from noise as a regularizer is to improve the stability of training. In the simple function computation networks, training was stable and fast across all training noise levels, and the main noise level we used (0.1) was selected without tuning. However, for the maze navigation network, we found that the training noise level we used represents an approximate lower bound for what is required to permit stable training. For lower noise levels, we observed issues with dynamical instability, resulting in frequent and catastrophic explosions in the performance cost. We observed similar instability in the multi-cognitive task suite network, which we initially tried to train with a noise level half that presented here (note that Driscoll et al. (2024) used lower training noise, but they trained for vastly longer, and used more tasks, which may have had a stabilizing effect on training).

## 4.2 Relevance of the noise preference phenomenon

We observed nonzero noise preferences in recurrent (Figures 2a-b, 3b-d, 6) and feed-forward (Figure 3d) noise-in networks with saturating activation functions, both rectifying and non-rectifying (Figure 6). Though our explanation for the emergence of a noise preference, which we established in Sections 3.3-3.5, caters to the relatively restrictive case of recurrent networks that depend on latent fixed points for computation, the essential elements of the explanation suggest the phenomenon is more general, as do our experiments with feedforward multilayer perceptrons (as opposed to RNNs) and the path integration task (which is a continuous attractor task rather than a fixed-point task). The essential elements of our explanation are the following. 1) When network performance is quantified via error between outputs and deterministic targets, networks are incentivized to reduce the variance of their outputs, and by extension, any latents on which the outputs depend. 2) Saturating activation functions, $output = f(input)$, induce noise-level-dependent activation (output) biases for noisy input distributions if a significant fraction of the input probability mass lies beyond a saturation boundary. They do this by asymmetrically attenuating the influence of the input noise on the downstream output noise. Changing the variance of the input distribution shifts the output distribution, because increased probability mass in only one tail of the input distribution has an influence on the output distribution. This is in contrast to when the input distribution is in a linear region of the activation function domain, where both tails of the input distribution influence the output equally, so the mean of the output does not depend on the dispersion of the input. 3) When the input distribution straddles a saturation boundary, the output distribution has lower variance than when the input distribution is in a linear region of the activation function.

Taken together, these key facts imply that saturating, noise-in networks trained to match their outputs to targets are incentivized to operate neurons near saturation as a way of reducing their output variance,

and therefore, they are incentivized to operate in a way that makes the means of their output distributions dependent on their synaptic noise. Noise preference develops because networks learn to compensate for this noise-dependence in their outputs, but their compensation is only correct when the synaptic noise is similar to what they saw in training. This explanation does not apply to noise-out networks because the second and third key facts listed above do not apply to noise-out networks, but it does apply to many networks of interest to neural network research in general. We framed our exploration in the context of computational neuroscience, but this was not necessary.

### 4.3 Relation to phenomena like stochastic resonance

We have presented a phenomenon involving an optimal, nonzero noise level, but the phenomenon we outline here is distinct from the well-known stochastic resonance (McDonnell & Abbott, 2009), and other similar phenomena such as coherence resonance (Pikovsky & Kurths, 1997; Hutt et al., 2020), and recurrence resonance (Katada & Nishimura, 2009; Metzner et al., 2024; Krauss et al., 2019). Speaking broadly, these phenomena apply to scenarios where a system's behavior can be described in terms of transitions between macro-states that are separated by thresholds (e.g., distinct attractors, regimes on either side of a bifurcation, half-spaces on either side of a detection threshold, etc.), and the performance criterion depends (either implicitly or explicitly) on both a measure of the ease with which the system makes transitions between macro-states and some measure of predictability. The role of the noise in these systems is to provide a constant stream of disturbances that can nudge the system across transition thresholds, and the existence of an optimal nonzero noise level stems from the tension between ease of transitions and predictability of the system: in the simplest version of stochastic resonance, detection of a weak sub-threshold signal is enabled by the addition of noise. Although our phenomenon arises from a vaguely similar tension (the networks learn to optimally trade neural coupling for a reduction in neural variance by operating neurons near saturation), the optimality of the noise for our networks is a byproduct of the optimality of our networks for the noise: that is, it is an overfitting phenomenon. In contrast, the resonance phenomena only occur in systems that are overall suboptimal for the performance objective, in that one could improve performance in a system exhibiting a resonance phenomenon by altering system parameters other than the noise level (to our knowledge this has only been explicitly discussed in the literature for stochastic resonance (Tougaard, 2000), but it is likely also the case for the other resonance phenomena mentioned above). Interestingly, with the inclusion of additional costs related to effort, e.g., metabolic cost, stochastic resonance phenomena can allow systems to use less energy by letting noise do some of the work in pushing the system state across transition thresholds, as shown in Koren et al. (2025). In this case, our phenomenon could increase efficiency the same way stochastic resonance does, as long as the effort cost does not directly penalize the injected noise (in Koren et al. (2025), the noise acted at the level of synaptic currents, but the metabolic cost penalized output spike trains, thus satisfying the requirement that the effort cost does not penalize the noise directly).

### 4.4 Future work

We have shown that a noise preference emerges in noise-in networks trained on a few different tasks, but all of our tasks, other than the path integration task, hinge on stabilizing network outputs around constant setpoints. As a result, most of our networks ended up relying on point attractors (Sussillo & Barak, 2013) to implement their computations, and the path integration networks likely relied on higher (two) dimensional continuous attractors (Khona & Fiete, 2022). However, many tasks relevant in neuroscience and beyond require computations that cannot be implemented by attractors with neutral stability on the manifold. One example is locomotion tasks, where the outputs should exhibit the characteristics of stable periodic motion (Seethapathi & Srinivasan, 2019; Laszlo et al., 1996). Such tasks lend themselves to computation via limit cycles, which require nonzero autonomous dynamics on the attractor manifold. In such instances, given the relatively permissive conditions that we have outlined here, we hypothesize that noise preferences may emerge. Future work should examine an even wider range of such tasks not easily solved by neutrally stable attractor dynamics, and ideally, also provide a mechanistic interpretation of the resulting phenomena.

Our numerical experiments used additive Gaussian noise. Although additive Gaussian noise is common in the literature (e.g., Yang et al. (2019); Driscoll et al. (2024)), explicitly signal-dependent noise models, such as multiplicative noise or Poisson noise, may be an important aspect of the noise environment of biological

systems (Jones et al., 2002; Tolhurst et al., 1983; Medina, 2011). Our explanation for the noise preference phenomenon depends on an induced signal dependence in the noise distribution: for neurons near saturation, the portion of the noise attenuated by the activation function depends on the net synaptic drive. So, other signal-dependent noise models could also result in a preference for specific noise parameters. Indeed, in the standard implementation of dropout (Srivastava et al., 2014), perhaps one of the most widely used signal-dependent noising procedures in the literature, a necessary step is to weight the layer outputs by the dropout probabilities, so that the dropout fraction can be set to zero at test time without degrading performance. Future work should investigate the potential for noise preference phenomena when using a variety of signal-dependent noise models.

Though we have established that noise can serve a purpose beyond regularization in noise-in networks, and we have drawn a distinction between noise-in and noise-out networks with respect to this point, we have not addressed how these two noising schemes differ in their effectiveness as regularization. Future work should involve some quantification of regularization as a function of noise level, both noise-in and noise-out. This would help in providing implications of our work for practitioners outside computational neuroscience, who may only be interested in noise as regularization, rather than as a necessity for the fidelity of their models.

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

## A  Training details

Code implementing all the experiments are provided as Supplementary Material as well as in this github repository: `https://github.com/manojsrinivasan/noisePreferenceInRNNs`. All training was done in MATLAB using the ADAM optimization algorithm (Kingma, 2014) with learning rate decay. For the function computation and single-neuron regulator networks, we used an initial learning rate of 0.001, a momentum weight of 0.9, an RMSprop weight of 0.999, and $\epsilon = 10^{-7}$. The learning rate half-lives were 2,500 batches for the function computation networks and 4,545 batches for the single-neuron networks; both were trained for a total of 10,000 batches. For the function computation networks, we used a batch size of 32 trials, and for the single-neuron networks, we used a batch size of one. There was no explicit regularization in these networks except for noise injection, either inside or outside the activation function for the function computation networks, and both inside and outside in different proportions for the single-neuron networks. For the function computation networks, we used a fixed noise standard deviation of 0.1, regardless of whether the noise was injected inside or outside the activation function. To initialize networks other than single-neuron regulator networks, we set all biases to zero and we set all weights to be normally distributed with zero mean and standard deviation equal to $0.8/\sqrt{\# \, of \, neurons}$ (Driscoll et al., 2024; Yang et al., 2019). For the function computation networks, we used 100 neurons. For the single-neuron regulator networks, the only trained parameter was the bias, and we initialized it to be equal to $r(1-w)$, which places the fixed point of the deterministic dynamics (of which there was only one, because all sampled weights were negative) exactly equal to the setpoint $r$. The initial neural activation states of the function computation networks were trained parameters set to zero at the start of training. For the regulator networks, the initial neural state was equal to the setpoint.

For the maze navigation network, we used an initial learning rate of 0.004 and a learning rate half-life of 1,500 batches, with the other ADAM parameters the same as for the function computation and single-neuron regulator networks. The maze network was larger than the function computation networks, with 672 neurons, but we trained for fewer batches, only 6,000 (this specific network size is not necessary for the noise preference phenomenon; it was inherited from unrelated work involving spatial embedding of neurons, for which this size was convenient). The batch size was 36, with any combination of starting and ending vertices equally likely and heterogeneous within a batch. Unlike for the other networks, we used some explicit regularization on the square of the maximum singular value of the recurrent weight matrix to protect against dynamical and numerical instability during training (e.g., Goudar et al. (2023)). We implemented this by adding the following term to the loss function:

$$\left(\max \frac{\|W_{\text{rec}} v\|}{\|v\|}\right)^2 \tag{6}$$

We added this term to the loss function with a scaling coefficient of 0.001. In addition to this explicit regularization, we also injected Gaussian noise inside the activation function with zero mean and standard deviation equal to 0.0894. The maze network had multiple trainable initial neural states (one for each vertex in the maze, with the initial state selected based on which vertex the trial starts on), each of which was set to the zero vector at the start of training.

For the multi-cognitive task suite networks, we tried to follow the training procedure outlined in Driscoll et al. (2024), with the following deviations:

- Instead of using the full list of 20 tasks, we use only ReactPro/ReactAnti, DelayPro/DelayAnti, and MemoryPro/MemoryAnti.

- We sampled the MemoryPro/MemoryAnti tasks twice as frequently as the others.

- We trained for only 15,000 batches.

- We used learning rate decay with a half-life of 7,500 batches.

- We only tried the ReLU activation function.

- We only tried an inside-noise level of 0.1, as for most of our other networks.

All networks except the tanh single neuron regulators used ReLU or the SoftPlus activation function. The SoftPlus activation function is defined as:

$$\text{SoftPlus}(\cdot) = \ln(1 + e^{\alpha(\cdot)})/\alpha, \tag{7}$$

where $\alpha$ is a scaling factor that controls the sharpness of the transition between the flat and sloped half-spaces. As $\alpha$ goes to infinity, the SoftPlus function approaches the ReLU function. For all networks in this paper, unless otherwise specified, we used $\alpha = 10$.

For the feed-forward multilayer perceptron we trained for function computation, we used the Adam optimization algorithm with learning rate decay. We used an initial learning rate of 0.001, a momentum weight of 0.9, an RMSprop weight of 0.999, and $\epsilon = 10^{-7}$. The learning rate half-life was 10,000 batches, and we trained for a total of 20,000 batches. We injected Gaussian noise inside the activation function (which we chose to be the ReLU activation function) with a standard deviation of 0.1. The cost function was the sum of a performance cost, which was just mean squared error, and an explicit regularization cost, which was the mean squared value of all the network weights. We added this to the cost with a multiplier of 0.001. The network had two hidden layers of width 100. The input and output layers both had only one unit, reflecting the fact that the sine function is univariate.

For the path integration task, we used the Adam optimization algorithm with learning rate decay. We used an initial learning rate of 0.001, a momentum weight of 0.9, an RMSprop weight of 0.999, and $\epsilon = 10^{-7}$. The learning rate half-life was 5,000 batches, and we trained for a total of 10,000 batches of size 32. We injected Gaussian noise inside the activation function (which we chose to be the ReLU activation function) with a standard deviation of 0.1. We used no explicit regularization, and the performance cost was simply RMSE between the actual particle position and the network output.

## B Task details

### B.1 Function computation

In the function computation tasks, we trained the networks to compute either the sine function over the interval $[0, 2\pi]$ or the hyperbolic tangent function over the interval $[-4, 4]$. In either case, the networks receive a one-dimensional input corresponding to the argument of the function being approximated. For the recurrent networks, their input remains on throughout the entire trial, so there was no requirement for the networks to "remember" the input. All trials lasted a total of 100 timesteps (feed-forward networks, of course, had no timesteps). In the deterministic readout variant of this task paradigm, the network output was read in the final time step for comparison to the target value; in the random readout variant, the readout could be read at any time step in the interval $[70, 100]$ with equal probability, forcing the network to maintain the correct output over that entire interval. In implementing the random readout version of the task, we found that the networks learned more quickly if we considered two readout times, one in the final time step and one randomly distributed over a fraction of the total duration, as previously described. The loss function for these tasks was the root mean squared error between the target values and the network outputs at readout times.

### B.2 Maze navigation

In the maze navigation task, we trained the network to navigate between any pair of vertices in a maze like the one shown in Figure 3a, pausing on vertices along the way. The network needed to pause on each vertex for a uniformly random number of timesteps, forcing it to rely on its inputs to determine when it should fixate at a maze vertex or move to the next. We loosely modeled the maze navigation framework after the multitask frameworks of Driscoll et al. (2024) and Yang et al. (2019) in the computational neuroscience literature. The network received three kinds of input: particle position state feedback, the destination indicator, and the fixation indicator. The particle state feedback was a 2-dimensional vector reporting the Cartesian coordinates of the particle in space in the current timestep. The destination indicator was a 2-dimensional vector indicating the Cartesian coordinates of the final destination vertex in space, and the fixation indicator was a 2-dimensional one-hot vector with both elements set to zero unless the network should be paused at a vertex. In timesteps when the network should be paused on the starting vertex of a trial, the first element of the fixation vector was set to one, and in timesteps when the network should be paused at any other vertex along the route, the second element was set to one. The network output was the instantaneous velocity of the particle, represented in 2-dimensional Cartesian coordinates. To force the network to learn to stabilize the particle along the maze routes, zero-mean Gaussian noise with variance equal to 0.0005 in each dimension was added to the network output in each time step, making it such that the particle would do a random walk if the network output were zero.

At the start of a trial, the particle position is initialized at one of the vertices, and the network state is initialized accordingly (see Section A). In the first 20 timesteps, a period we refer to as the downtime period, the second element of the fixation vector is on, and the destination indicator shows the coordinates of the starting maze vertex, making the input conditions identical to if we had just finished a route terminating at the current starting vertex. Then, over a period with length uniformly distributed in $[30, 50]$ timesteps, which we call the starting fixation period, only the first element of the fixation vector is on, and the destination vector shows the coordinates of the destination for the present trial, which is how it will remain for the remainder of the trial. During the downtime and starting fixation periods, the network must remain at the starting maze vertex. The rest of the trial then alternates between jaunt periods, intervals 20 timesteps long when both elements of the fixation vector are set to zero and the network must navigate the particle to the next vertex along the route, and via fixation periods, which have length distribution identical to that of the starting fixation period, and during which only the second element of the fixation vector is turned on. For computational efficiency, we buffered out all trials to be 280 steps long, regardless of how long their corresponding routes were. To do this, we added extra time steps to one of the fixation periods at random, so there was a chance that any one pause could exceed the intervals previously given. The loss function for the maze task was the root mean squared error between the particle position and its target position over all timesteps and trials in a batch. The target position for the particle during pauses was simply the

vertex position at which it was supposed to be pausing, and during jaunt periods, we used a smooth cubic polynomial interpolator between the starting and ending vertices of the jaunt, constraining the starting and ending velocities of the jaunt to be zero.

### B.3 Multi-cognitive task suite

We implemented only the ReactPro/ReactAnti, DelayPro/DelayAnti, and MemoryPro/MemoryAnti tasks from Yang et al. (2019). Instead of the more biological versions used in Yang et al. (2019), which encode stimuli and responses as activity bumps on rings of input/output units, we opted to use the versions from Driscoll et al. (2024), which encode the directional stimuli/responses with 2D unit vectors. We also only use a single stimulus modality (rather than two, as done in Yang et al. (2019) and Driscoll et al. (2024)).

### B.4 Path integration

We roughly recreated the path integration task exemplified in Cueva & Wei (2018). The network received inputs of particle heading (in radians) and discrete speed (spatial length units), and it had to use these to update its outputs to reflect the current particle position. The particle's motion was totally autonomous, so the network was simply responding to its motion with no ability to affect it and no need to distinguish re-afferent from ex-afferent feedback. The particle executed a smoothed random walk of approximately constant speed (heading changes in each time step were sampled from a normal distribution with $\mu = 0°$ and $\sigma = 20°$, and the step lengths were sampled from a normal distribution with $\mu = 0.1$ units and $\sigma = 0.01$ units). If this random walk brought the particle outside the 2.5 unit x 2.5 unit box in which the particle was constrained, the step was rejected, and the velocity was corrected as if the particle had elastically collided with the wall. Each trial began with the particle in the center of the box and a uniformly random heading, and trials lasted 100 time steps, typically resulting in a few ($< 10$) wall collisions.

### B.5 Single-neuron regulator

The scalar neural state $h_t$ of the single-neuron CTRNN with noise both inside and outside its activation function has dynamics defined by the update rule:

$$h_{t+1} = (1 - \gamma)h_t + \gamma(f(wh_t + b + \mathcal{N}(0, \sigma_{\text{in}}^2)) + \mathcal{N}(0, \sigma_{\text{out}}^2)), \tag{8}$$

where $h_1 = r$ and $f(\cdot)$ is either the ReLU function or the tanh function. We considered the regulator problem (Anderson & Moore, 2007) described by the loss function:

$$\min \frac{1}{T} \sum_{t=1}^{T} (h_t - r)^2, \tag{9}$$

where $r \geq 0$ is the constant setpoint for $h_t$. We fixed $\sigma_{\text{out}}$ (the noise standard deviation outside the activation function, set to one), $T$ (the number of simulation time steps, set to 50) and $\gamma$ (the ratio of the simulation timestep to the neural time constant, set to 0.2), and found the optimal value of $b$ (the activation bias) for different combinations of $w$ (the recurrent weight), $\sigma_{\text{in}}$ (the noise level inside the activation function), and $r$ (the setpoint). We then measured the difference in performance between when $\sigma_{\text{in}}$ is left at the value for which the networks were optimized and when it is set to zero. For each combination of $w$ and $\sigma_{\text{in}}$, we also identify the value of the setpoint $r$ that permits the lowest loss function value among those we sampled. We considered a fine sampling of 100 values for both $r$ (evenly spaced in the interval $[0,1]$) and $\sigma_{\text{in}}$ (evenly spaced in the interval $[0,2]$), and we considered four values of $w$, namely $\{-1, -2, -3, -4\}$, with the largest magnitude $w = -4$ selected such that the deterministic dynamics would extinguish any deviation from the fixed point in a single time step if there were no noise. We set the post-activation noise level to one for all experiments while varying the setpoint location and the inside noise level, without loss of generality, because the fixed point shift phenomenon depends on how far a fixed point is from nonlinear boundaries *in comparison to* the noise standard deviation.

## C   Fixed point finding with and without noise

To find the fixed points of the noise-in function computation networks without accounting for noise-induced shifts  (Driscoll et al., 2024), we optimized the following objective function with respect to the neural state $h$:

$$\|\gamma(-h + f(W_{\text{rec}}h + W_{\text{in}}u + B_{\text{in}}))\|^2. \tag{10}$$

This objective function is simply the squared magnitude of the discrete change in $h$ if we take a single simulation step starting from $h$, so its local minima correspond to locations in the neural activation space where the network state changes most slowly in a local neighborhood. We optimized this objective using ADAM with learning rate decay, using an initial learning rate of 0.07, a momentum weight of 0.9, an RMSprop weight of 0.999, $\epsilon = 10^{-7}$, and a learning rate half-life of 1,386 optimization steps. We terminated the optimization when the objective dropped below $4.9 \times 10^{-9}$. To find the fixed point corresponding to a particular argument of the function to be approximated by the network, we ran the network without noise over a full trial with that argument, taking the final neural state as the initial guess for the fixed point, and then we optimized the objective function with $u$ set to the argument of interest.

To find the fixed points of the noise-in function computation networks with noise accounted for, we followed the same procedure as for the noise-free fixed points, but using the following cost:

$$\|\gamma(-h + E(f(W_{\text{rec}}h + W_{\text{in}}u + B_{\text{in}} + \mathcal{N}(0, \sigma^2))))\|^2, \tag{11}$$

where $E(\cdot)$ denotes a Monte Carlo approximation of the expected value, evaluated by averaging over 300 samples, and $\sigma$ is the noise standard deviation used during network training, equal to 0.1. For these optimizations, we used the same optimization parameters as for the noiseless procedure, except we used an initial learning rate of 0.0175, and we terminated the optimization when the objective function dropped below $\frac{1}{100}\sigma^2(\# \text{ of neurons})/(\# \text{ of noisy samples}) = \frac{1}{3} \times 10^{-4}$.

To compute the magnitude of the output fixed point shift induced by the noise, we simply took the difference between the noisy fixed point and the noise-free fixed point corresponding to each input argument, left-multiplied it by the output matrix $W_{\text{out}}$, and took its magnitude.

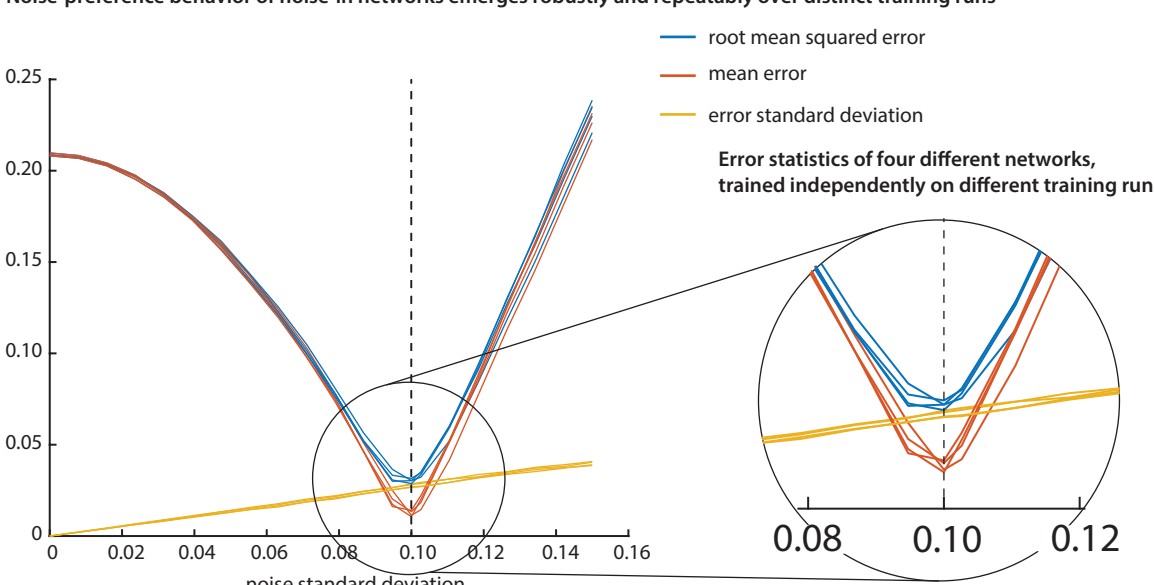

Figure A1: **Noise preference across training runs.** Illustrative example showing the noise-preference behavior was qualitatively similar for networks obtained from different training runs, each starting from different initial network initializations and trained with different seeds for noise realizations. Noise-preference error statistics curves from four different training runs are shown for the function computation task; not only are the curves qualitatively similar with a nonzero noise preference, they are also quantitatively similar, with some small variability to be expected from the stochasticity of the training process and finiteness of training runs. This robustness to training runs was generally true across the different task examples we considered (as long as the training noise was large enough as indicated in Figure 2b).

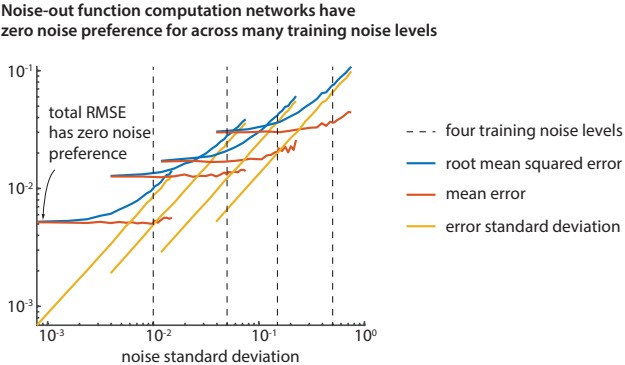

Figure A2: **Noise-out networks with multiple training noise levels.** Noise-out networks did not develop a non-zero noise preference for a range of training noise levels across multiple orders of magnitude. Four training noise levels and the corresponding errors and error components are shown. This figure is the noise-out analog of Figure 2b, which showed that the noise-in networks did develop a non-zero noise preference as long as the training noise level was high enough.

