# OpenReview forum: "Paradoxical noise preference in RNNs"
_TMLR — Accepted by TMLR_

### Review · Reviewer_k7Bq · 2026-02-04

**Summary Of Contributions:**

The work studies the sensitivity of continuous-time recurrent neural networks to Gaussian white noise added either inside or outside of the activation function, during training and at test. They describe a phenomenon that is counter to the common understanding of noise during training serving a regularising function. Under that interpretation, noise added during training can improve performance at noise-free test time. Contrary to this understanding, the authors report observations of networks that strongly prefer at test time the same noise level experienced during training. However, this occurs only when the noise is added inside the activation function (noise-in), not outside (noise-out). This phenomenon is studied in three supervised learning tasks: function approximation, navigation, and single neuron regulation. The authors offer an explanation for this phenomenon based on an analysis of how noise can shift the fixed points of the learning dynamics. This shift can occur both in noise-in and noise-out networks but is more likely to occur in noise-in networks which have more fixed points near boundaries between linear regions of their dynamics. Ultimately, the noise preference observed is an over-fitting phenomenon.

### Strengths:

- The figures are beautiful and impressively clear
- The paper is well-written and easy to read, especially the Introduction, Methods, and Results
- A very specific phenomenon is carefully studied in multiple task settings

### Weaknesses:

- The ‘why’ is missing. It is not clear who the audience for this paper is and why they would care about the result. The authors mention in the abstract and introduction that recurrent neural networks are central in computational neuroscience, but never describe the relevance of the particular claims made here for neuroscience
- Only a single activation function, rectified linear, is considered.

**Audience:**

Yes

**Audience Explanation:**

The findings of this paper will be potentially interesting to those who want to understand the mechanisms of recurrent neural computation.

**Broader Impact Concerns:**

I see no ethical implications to be considered in this primarily theoretical work.

**Claims And Evidence:**

Yes

**Claims Explanation:**

The main observations that constitute the phenomenon and the authors predictions about the relationship between error and noise-induced changes in fixed points are clearly visible in the figures. Noise preferences are studied in three simple and complementary tasks, providing converging evidence. Results for the noise-out networks are only included for the function approximation task.

**Requested Changes:**

### Critical
- Please discuss why your findings matter and to whom. You mentioned neuroscience in the introduction. Do your results suggest any hypotheses about neural computation? Or have any consequences for computational neuroscientists or machine learning researchers? What broader scientific goal is this work contributing to? The results are very clear but why you did this analysis is less so. Please contextualize.
- As far as I can tell, all experiments/analysis are performed with a rectified linear nonlinearity. It is not clear how much of the results depend on this particular activation function and its sharp cutoff at 0 (and the fact that the added noise is 0 mean). Given the role that the saturation of the activation function plays in your explanation of when noise preferences emerge, I suggest including supplemental figures for e.g., a sigmoid activation function and making more explicit what aspects of the results are general for any nonlinearity and which are specific for rectified linear.

### Suggested (not critical for acceptance)
- The performance of the noise-in and noise-out networks are never directly compared to each other or to networks without noise added during training. Do we see the regularizing effect described in the introduction for the noise-out networks? Can we only say that noise-in networks develop a noise preference or can we additionally say that the noise-out networks generalise better than noise-free networks? Do the noise-in networks perform worse even at their preferred noise-level? An additional analysis and figure answering these questions would strengthen the paper but is not necessary to support the main claims.

---

> ### Author Response · Authors · 2026-03-03
> **Response to reviewer k7Bq (concerns 1 & 2)**
>
> We thank the reviewer for the clear and thoughtful summary of our work and for recognizing the noise-preference phenomenon, our fixed-point–based explanation, and the careful evaluation across multiple tasks. We are grateful for the positive feedback on the clarity of the writing and figures, and we appreciate the reviewer’s constructive assessment of the paper’s strengths. We address your helpful constructive comments below:
>
> **Concern 1. Explain why this phenomenon is important and to whom.**
>
> > "The ‘why’ is missing. It is not clear who the audience for this paper is and why they would care about the result. The authors mention in the abstract and introduction that recurrent neural networks are central in computational neuroscience, but never describe the relevance of the particular claims made here for neuroscience." "Please discuss why your findings matter and to whom. You mentioned neuroscience in the introduction. Do your results suggest any hypotheses about neural computation? Or have any consequences for computational neuroscientists or machine learning researchers? What broader scientific goal is this work contributing to? The results are very clear but why you did this analysis is less so. Please contextualize."
>
> **Response:** The following paragraphs in the manuscript explain the broader significance of the work to computational neuroscientists:
> - We have shown in **section 3.2** how the noise preference phenomenon occur in a number of models of relevance in computational neuroscience, especially, maze-following networks and multi-task networks. This section has now been expanded to provided additional examples of relevance to computational neuroscience, specifically, the multi-task networks performing a suite of cognitive tasks relevant to computational neuroscience (new **section 2.2.3**, additions in **section 3.2**, and new section in **appendix B.3**, new figure panel, **figure 3c**).
>
> - As the second paragraph of the Discussion section, we have added the following text, addressing implications for different audiences: _"Our findings demonstrate that training noise can become an integral component of the computation learned by a network, shaping fixed-point structure and inducing systematic output biases that render performance intrinsically dependent on the training noise level rather than merely regularized by it. For computational neuroscience, this suggests a concrete hypothesis: neural circuits operating near nonlinear activation boundaries may leverage—or become functionally dependent on—synaptic variability to stabilize activity and reduce variance, implying that some observed noise sensitivity in biological systems could reflect learned dynamical structure rather than incidental variability. For machine learning researchers, the results highlight a subtle failure mode of noise injection, namely overfitting to the stochastic training environment itself, and thereby contribute to the broader scientific objective of understanding how dynamical systems perform computation under noise, clarifying when variability serves as a stabilizer or regularizer versus a hidden dependency."_
>
> - In the Discussion, specifically **section 4.3**, we have added commentary making connections to to other recent work (Koren et al, 2025) that showed noise may reduce energy expenditure in some situations.
>
> **Concern 2. Also consider other activation functions.**
> > "Only a single activation function, rectified linear, is considered." "As far as I can tell, all experiments/analysis are performed with a rectified linear nonlinearity. It is not clear how much of the results depend on this particular activation function and its sharp cutoff at 0 (and the fact that the added noise is 0 mean). Given the role that the saturation of the activation function plays in your explanation of when noise preferences emerge, I suggest including supplemental figures ..."
>
> **Response:**
> - Good point. We agree that treatment of a nonrectifying activation function is useful, as our proposed explanation for the noise preference phenomenon should apply to noise-in networks with _any_ saturating activation function. We have addressed this by running new experiments with our single neuron regulator network with tanh activation (instead of reLU activation). These new results are reported in the **new figure 6b**.
> - We have added corresponding new text in **Section 3.5** on the **single-neuron regulator network** _".. Further, this reasoning does not require that the activation function be rectifying, simply that it should be saturating, which is supported by the presence of the phenomenon in both ReLU and tanh single-neuron regulator experiments (Figure \ref{singleNeurRegResultsFig}a-b). "_
> - New text in Figure 6b caption: _"b) Using tanh activation function also results in noise preference: because tanh saturates for both high and low input values, we correspondingly find noise preference when the setpoint location is high or low."_

---

> > ### Author Response · Authors · 2026-03-03
> > **Response to reviewer k7Bq (concerns 3)**
> >
> > **Concern 3 (not critical). Noise-in versus noise-out networks**
> >
> > > Suggested (not critical for acceptance): “The performance of the noise-in and noise-out networks are never directly compared to each other or to networks without noise added during training. Do we see the regularizing effect described in the introduction for the noise-out networks? Can we only say that noise-in networks develop a noise preference or can we additionally say that the noise-out networks generalise better than noise-free networks? Do the noise-in networks perform worse even at their preferred noise-level? An additional analysis and figure answering these questions would strengthen the paper but is not necessary to support the main claims.”
> >
> > **Response:** We have added the following text to the Discussion **section 4.4** to address these questions as potential future work: _"Though we have established that noise can serve a purpose beyond regularization in noise-in networks, and we have drawn a distinction between noise-in and noise-out networks with respect to this point, we have not addressed how these two noising schemes differ in their effectiveness as regularization. Future work should involve some quantification of regularization as a function of noise level, both noise-in and noise-out. This would help in providing implications of our work for practitioners outside computational neuroscience, who may only be interested in noise as regularization, rather than as a necessity for the fidelity of their models."_
> >
> > We have also added additional experiments at various training noise levels, which support some brief comments about the regularizing potential of noise injected inside the activation function (though we make no comparison to noise injected outside the activation function). See new **figure 2b** and **section 4.1**.

---

> > ### Comment · Reviewer_k7Bq · 2026-03-09
> >
> > These additions perfectly address my concerns. The text you've added to the Discussion makes it much more explicit what the implications of these findings are. The tanh experiments and and associated explanation helps the reader to understand the generality of the findings.

---

> > > ### Author Response · Authors · 2026-03-15
> > > **Second response to reviewer k7Bq**
> > >
> > > We are happy to have addressed all your concerns, and we thank you for all your help in making the manuscript better. Thank you also for your kind note that it is okay to limit our work to fixed point task, as long as we have acknowledged that scope clearly. Ultimately, we decided to add some non-fixed-point-task experiments, specifically involving path integration, to address comments by reviewer MExH.

---

### Review · Reviewer_Y2Tt · 2026-02-05

**Summary Of Contributions:**

The paper addresses the role of the noise in continuous-time recurrent neural networks. It finds that if noise is added inside the neural activation function, the network will optimise its test performance in the presence of the noise at the level of the training noise. If the noise was added outside of the activation function during training, the test performance is optimal with zero noise level. The effect of the noise on network performance is addressed with several tasks with different levels of complexity. Converging results from these different tasks suggest that the conclusion about the role of the noise in noise-in and noise-out networks is independent of the specific task that the network has to solve.

**Audience:**

Yes

**Audience Explanation:**

The results of the paper are interesting from both methodological as well as neurobiological perspective. From the methodological perspective, it is useful to know when the noise is helpful to establish better training methods. From neurobiological perspective, a potential utility of the noise is also very interesting and intriguing. If the neural noise such as synaptic noise is unavoidable in a biological system, it would be very interesting to learn that such "necessary evil" can be turned into a process that benefits network function.

**Broader Impact Concerns:**

I do not see any concerns on the ethical implications of the work.

**Claims And Evidence:**

Yes

**Claims Explanation:**

The paper addresses the question of when the noise is useful to continuous-time RNNs in a way that is, for most of it, clear and transparently. Authors perform training on tasks of various complexity, which helps the paper to go beyond a specific case scenario and justifies the general conclusion.

**Requested Changes:**

The paper would benefit addressing and clarifying a couple of questions.

1) Results are relevant for continuous-time RNNs. Why is it important that the networks operates in continuous time? Would networks that do not operate in time domain show a noise preference after training if the noise was inside the activation function? Authors could discuss this to better frame their results and strengthen their contribution.

2) When the noise is injected outside of the activation function, the noise still shifts the fixed points. Why then the noise has no impact on the test performance of the network? I see this as a main caveat of the paper, clarifying it is critical for my recommendation.

3) In Equation (4), authors define a piecewise linear differential equation that does not make much sense to me. Is there a typo? Should it be "...with $x<0$" in the second line? Can authors please check the equation and revise if necessary?

4) From the caption of the Figure 4A I expect an inset, but it seems that it has been forgotten. Authors should add the inset to strengthen the coherence of the text and thus strengthen the work.

5) A recent study [1] on efficient and biologically plausible recurrent spiking networks found that intermediate levels of noise benefit the (metabolic) efficiency of the networks. They implemented the noise as a Gaussian noise in the update equation for the membrane potential, which is similar to the Eq. 2-3 in your contribution. Could authors comment about how their results relate to this study? This would help to strengthen the work as it would better situate the current contribution among the recent computational / systems neuroscience literature.

[1] Koren, V., Malerba, S. B., Schwalger, T., & Panzeri, S. (2025). Efficient coding in biophysically realistic excitatory-inhibitory spiking networks. Elife, 13, RP99545.

---

> ### Author Response · Authors · 2026-03-03
> **Response to reviewer Y2Tt (Concern 1)**
>
> Thank you for the clear summary of our contributions and for noting that our claims are supported by mostly clear and transparent evidence across multiple tasks of varying complexity. We appreciate the recognition of the work’s potential interest to the TMLR audience from both methodological and neurobiological perspectives.
>
> **Concern 1: why continuous-time RNNs?**
> > Results are relevant for continuous-time RNNs. Why is it important that the networks operates in continuous time? Would networks that do not operate in time domain show a noise preference after training if the noise was inside the activation function? Authors could discuss this to better frame their results and strengthen their contribution.
>
> **Response:** Thank you for this question. We agree that some discussion of whether we expect the phenomenon in networks that don’t explicitly operate in time would strengthen our contribution. We do expect the phenomenon to persist when the network does not operate in continuous time. e.g., feedforward networks. To address your comment, we’ve performed new experiments with a simple feed-forward MLP and added a new figure panel with these results, specifically **figure 3d**.
>
> - The caption for this new figure panel says: _d) Even feedforward multilayer perceptrons trained on the function computation task exhibit a noise preference, demonstrating that this phenomenon is relevant outside the context of time-evolving networks._
>
> - _We also trained simple noise-in feedforward multilayer perceptrons (MLPs) on a function computation task (only the sine function, see Appendix \ref{trainDets} for network and training details) to see if the phenomenon is specific to recurrent networks, and we found that a noise preference, driven by error bias, also emerges here (figure \ref{mazeFig}d)._
>
> - We have added a paragraph in the Appendix containing the methods for these new experiments.

---

> > ### Comment · Reviewer_Y2Tt · 2026-03-08
> >
> > I thank the authors for performing additional analysis on MLPs and thus broadening the generality of their results.

---

> ### Author Response · Authors · 2026-03-03
> **Response to reviewer Y2Tt (Concern 2)**
>
> **Concern 2: why no noise-preference for noise-out networks**
> > “When the noise is injected outside of the activation function, the noise still shifts the fixed points. Why then the noise has no impact on the test performance of the network? I see this as a main caveat of the paper, clarifying it is critical for my recommendation.”
>
> **Response:** The 'single-neuron regulator' in the original draft was indeed to elucidate this difference, now expanded.
>
> _Since both noise-in and noise-out networks can exhibit noise-dependent fixed point shifts, but we only see a noise preference in noise-in networks, we hypothesized that there may be some performance incentive specific to noise-in networks that drives the development of a preference. Given our reasoning in sections 3.3.1 and 3.3.2, we sought to understand how the loss function varies with the distance between fixed points and nonlinear boundaries in the dynamics. For simplicity, we use the single-neuron regulator task to investigate this._
>
> _The single-neuron regulator experiments showed that there are regions in the parameter space where networks (both ReLU and tanh) exhibit a noise preference. When the pre-activation noise dominates the post-activation noise, the best networks tend to land inside or near these regions with a noise-preference, but when the post-activation noise dominates, they land squarely outside these regions (Figure 6}a). We believe this trend is due to a tradeoff. If the stationary distribution straddles the saturation boundary, a portion of the pre-activation noise gets turned off, while leaving a portion of the post-activation noise out of reach for the recurrent weight to help in drawing the state back toward its expected value. For a network dominated by pre-activation noise, this is a good trade, so the stationary distribution should be placed near the boundary, but for a network dominated by post-activation noise, this is a bad trade, so the stationary distribution should be placed far from the boundary. This tracks with the observation that networks with higher magnitude recurrent weights tolerate higher inside noise levels before opting to place their setpoints near saturation boundaries. Given the reasoning in sections 3.3.1 and 3.3.2, this may explain why noise-in networks systematically develop a noise preference while noise-out networks do not. Further, this reasoning does not require that the activation function be rectifying, simply that it should be saturating, which is supported by the presence of the phenomenon in both ReLU and tanh single-neuron regulator experiments (Figure 6a-b)._
>
> We added the following text to **section 4.2** (discussion), expanding on the key intuition for the noise preference.
>
> _ The essential elements of our explanation are the following. 1) When network performance is quantified via error between outputs and deterministic targets, networks are incentivized to reduce the variance of their outputs, and by extension, any latents on which the outputs depend. 2) Saturating activation functions, $output=f(input)$, induce noise-level-dependent activation (output) biases for noisy input distributions if a significant fraction of the input probability mass lies beyond a saturation boundary. They do this by asymmetrically attenuating the influence of the input noise on the downstream output noise. Changing the variance of the input distribution shifts the output distribution, because increased probability mass in only one tail of the input distribution has an influence on the output distribution. This is in contrast to when the input distribution is in a linear region of the activation function domain, where both tails of the input distribution influence the output equally, so the mean of the output does not depend on the dispersion of the input. 3) When the input distribution straddles a saturation boundary, the output distribution has lower variance than when the input distribution is in a linear region of the activation function.
> Taken together, these key facts imply that saturating, noise-in networks trained to match their outputs to targets are incentivized to operate neurons near saturation as a way of reducing their output variance, and therefore, they are incentivized to operate in a way that makes the means of their output distributions dependent on their synaptic noise. Noise preference develops because networks learn to compensate for this noise-dependence in their outputs, but their compensation is only correct when the synaptic noise is similar to what they saw in training. This explanation does not apply to noise-out networks because the second and third key facts listed above do not apply to noise-out networks, but it does apply to many networks of interest to neural network research in general. We framed our exploration in the context of computational neuroscience, but this was not necessary._
>
> We have expanded a line in the abstract, drawing better attention to these results.

---

> > ### Comment · Reviewer_Y2Tt · 2026-03-08
> >
> > 1) Thank you for clarifying the mechanism of noise effect on performance in noise-in and noise-out networks. For me, the additions in result section do not explain the mechanism, while the additions in the discussion section do. As a suggestion, authors could consider distilling even better the text so that it renders the mechanism more clearly.
> >
> > 2) The claim "This tracks with the observation that networks with higher magnitude recurrent weights tolerate higher inside noise levels before opting to place their setpoints near saturation boundaries." is not obvious to me. Could authors explain the reasoning that would support that the magnitude of recurrent weights influences the optimal noise level?

---

> > > ### Author Response · Authors · 2026-03-15
> > > **Second response to reviewer Y2Tt**
> > >
> > > We are glad to have adequately addressed most of your concerns, and we thank you for insisting on a clearer explanation of the discrepancy between noise-in and noise-out networks in the results section (while you found the analogous parts in the Discussion section satisfactory). This is a critical point, and we agree we don’t want ineffective language to muddy the waters. We have rewritten the results paragraph that you found unclear, this time trying to be as clear as possible and incorporating some parallel language to the paragraph in the Discussion that you found more effective and satisfactory.  Specifically, we addressed your Point 1 by broadly reworking Section 3.5 paragraph 3, and, in that same section, we addressed Point 2 adding a clarifying clause at the end of the sentence that began: “This tracks with ...” and slightly rewording it (the sentence now starts with "This is consistent with ..."). Thanks again for the opportunity to make this clearer.

---

> > > > ### Comment · Reviewer_Y2Tt · 2026-03-17
> > > >
> > > > I thank the authors to improve text clarity on this important point.
> > > >
> > > > Authors have addressed all the weak points I raised. In light of these improvements and of improvements brought by addressing criticism of other reviewers, I recommend the paper for acceptance.

---

> ### Author Response · Authors · 2026-03-03
> **Response to reviewer Y2Tt (Concerns 3, 4, 5)**
>
> **Concern 3: Typo in equation**
> > In Equation (4), authors define a piecewise linear differential equation that does not make much sense to me. Is there a typo? Should it be "...with x<0" in the second line? Can authors please check the equation and revise if necessary?
>
> **Response:**
> Good catch, this was indeed a typo, and we’ve fixed it in the current revision. The upper case is <= 0 (first line) and the lower case is >0 (second line).
>
> **Concern 4: Missing inset?**
> > From the caption of the Figure 4A, I expect an inset, but it seems that it has been forgotten. Authors should add the inset to strengthen the coherence of the text and thus strengthen the work.
>
> **Response:** Another good catch: the text in the caption referred to an inset that shouldn’t have been there (we had removed an inset during an internal revision process before the first submission). We have now harmonized the figure caption with the figure, and removed the reference to any inset in figure 4a.
>
> **Concern 5: Optimal noise for metabolic efficiency in recent work **
> > A recent study [1] on efficient and biologically plausible recurrent spiking networks found that intermediate levels of noise benefit the (metabolic) efficiency of the networks. They implemented the noise as a Gaussian noise in the update equation for the membrane potential, which is similar to the Eq. 2-3 in your contribution. Could authors comment about how their results relate to this study? This would help to strengthen the work as it would better situate the current contribution among the recent computational / systems neuroscience literature.
>
> **Response:** We found this to be an interesting reference, and we agree with the authors' remark in that manuscript that this is a manifestation of 'stochastic resonance.' We have added the following discussion of this work to our discussion paragraph devoted to stochastic resonance-like phenomena:
>
> _Interestingly, with the inclusion of additional costs related to effort, e.g., metabolic cost, stochastic resonance phenomena can allow systems to use less energy by letting noise do some of the work in pushing the system state across transition thresholds, as shown in Koren et al, 2025. In this case, our phenomenon could increase efficiency the same way stochastic resonance does, as long as the effort cost does not directly penalize the injected noise (in Koren et al, 2025, the noise acted at the level of synaptic currents, but the metabolic cost penalized output spike trains, thus satisfying the requirement that the effort cost does not penalize the noise directly)._

---

> > ### Comment · Reviewer_Y2Tt · 2026-03-08
> >
> > I thank the authors for correcting the typos and the inset inconsistency, as well as for adding an interesting discussion point to their paper.

---

### Review · Reviewer_MExH · 2026-02-17

**Summary Of Contributions:**

**Summary:**

This paper uncovers and empirically analyses "noise level preferences" that arise in recurrent neural networks trained either with noise within the nonlinearity ("noise-in") or outside it ("noise-out"). The authors train noise-in and noise-out networks on some simplistic tasks such as function computation, maze navigation, and a single-neuron regulation task (with both noise-in and noise-out simultaneously). Through these experiments, the authors identify that noise-in networks develop a preference for the training noise level, requiring it at test time, while noise-out networks do not. This is explained as the effect of the amount of shift in the fixed points the network learns to solve the task; the authors find that the shift in fixed points is correlated with the error at zero noise for these noise-in networks. Overall, the paper reveals and attempts to explain an interesting behaviour of these RNNs that are widely used as models of neural computation.

**Strengths:**
* This noise-preference phenomenon and differences between noise-in and -out networks is a novel analysis to my knowledge.
* The paper is clearly presented with neat figures.
* While the tasks are simplistic, a good set of experiments have been presented to elucidate the issue, and the analyses such as correlating the zero noise error and fixed point shift are interesting and illuminating.

**Weaknesses:**
* The results in Section 3.1 and Figure 2 seem to be for only one training noise level. I would expect to see a sweep over several different noise levels and a heat map characterising the summary of trends for such experiments. Currently, it is difficult to understand what would happen at very low or very high noise levels since only one training noise value is presented. In particular, I would have liked to see some argument relating this effect and overall performance to the effective signal-to-noise ratio (i.e., how the scale of the noise vis-a-vis the scale of the inputs or preactivations, etc. affects both task performance and noise level preference).
* A key limitation, and this is acknowledged by the authors, is the focus on just tasks with fixed point dynamics. This is quite restrictive for a subset of tasks and there are several simplistic tasks that one could implement with RNNs, such as path integration tasks, which show more complex attractor dynamics (e.g., spatial navigation -> planar attractor, evidence integration -> line attractor, head direction -> ring attractor, etc.). It would be nice to see what happens in these tasks as well, and I would argue that these would be important to include for the paper to be complete + comprehensive.
* For "neuroscientifically-relevant multi-task networks", i.e., the results in Section 3.2, I would have expected to see more task diversity rather than just one maze task with different start/end configurations being different tasks. The cited papers such as Yang et al. (2019) and Driscoll et al. (2024) show clear examples of RNNs jointly trained on a wide variety of tasks. It would be nice to see experiments with actual multitask RNNs such as this (e.g., on the ModCog suite – Khona et al. (2022) – https://github.com/mikailkhona/Mod_Cog).
* This is relatively minor among my weaknesses because I appreciate empirical work, but it would also be nice to have a theoretical understanding of why this phenomenon occurs, e.g., by trying to understand why noise compensation is learnt in noise-out networks (see Krishna et al. (2024), for example) but not for noise-in networks, causing them to overfit to and rely on the training noise level.
* Another relatively minor point is that some neuroscientifically-relevant works such as Burak & Fiete (2012), Krishna et al. (2024), or Bredenberg et al. (2026) discuss noise-out networks unlike the others presented here. This could still be consistent with the rate-based interpretation of these models as long as the noise if Gaussian is smaller in scale than the firing rates themselves. In the first (Burak & Fiete, 2012), this view allows one to see memory drift as Brownian motion on a continuous attractor manifold. In the second (Krishna et al., 2024), noise is shown to be integral to the emergence of a neuroscientifically-relevant phenomenon (quiescent replay being a result of diffusion on the attractor manifold). The third (Bredenberg et al., 2026) shows denoising to be an important part of the computation and for improved quality of generated images. This explicitly relates to the point in the introduction that noise may have actual roles in computation itself (and not necessarily something to drown out), rather than being used in biological models solely to mimic synaptic noise or help alleviate training instabilities. It would be nice to add a discussion on this aspect and such works, especially given that a few of these works study tasks that would be relevant here (continuous attractor dynamics).

**References:**
* Yang et al. "Task representations in neural networks trained to perform many cognitive tasks." Nature neuroscience 22.2 (2019): 297-306.
* Driscoll et al. "Flexible multitask computation in recurrent networks utilizes shared dynamical motifs." Nature Neuroscience 27.7 (2024): 1349-1363.
* Krishna et al. "Sufficient conditions for offline reactivation in recurrent neural networks." The Twelfth International Conference on Learning Representations (2024).
* Burak, Yoram, and Ila R. Fiete. "Fundamental limits on persistent activity in networks of noisy neurons." Proceedings of the National Academy of Sciences 109.43 (2012): 17645-17650.
* Bredenberg, Colin, et al. "The oneirogen hypothesis: modeling the hallucinatory effects of classical psychedelics in terms of replay-dependent plasticity mechanisms." eLife 14 (2026).

**Additional Comments:**

It might be nice apart from just quantifying the shift in the fixed points to actually visualise the results of the fixed point finder algorithm.

**Audience:**

Yes

**Audience Explanation:**

Yes, this paper aligns with the following focus areas of TMLR:

* experimental and/or theoretical studies yielding new insight into the design and behaviour of learning in intelligent systems;
* accounts of applications of existing techniques that shed light on the strengths and weaknesses of the methods.

It would be of particular interest to those working on recurrent neural networks/sequence models in general, and computational neuroscientists.

**Broader Impact Concerns:**

No concerns.

**Claims And Evidence:**

Yes

**Claims Explanation:**

I have listed my weaknesses in the section above. I believe that the paper is very close to having enough support for its claims through experimental evidence, I just find the current breadth of tasks and sweeps over noise to be lacking for the paper to be considered complete and comprehensive. If the authors were to resolve my main weaknesses (non-fixed point tasks, more training noise values and sweeps over these values at training and test time, actual multi-task networks with task diversity), I would be happy to revisit my rating here.

Post-discussion edit: The authors addressed all key weaknesses that I had raised in my initial review and during the discussion period. I now believe the paper's claims are supported by clear, convincing evidence.

**Requested Changes:**

* Please add a sweep over different training noise levels in addition to different testing noise levels.
* Please add more tasks that do not just involve fixed point dynamics. Even simple continuous attractor tasks would in my opinion strengthen the paper (examples have been provided above).
* If possible, multi-task networks with more task diversity and not just different configurations of the same task would be nice to include.
* Please attempt to discuss the final point in my "weaknesses" about noise-out networks in computational neuroscience.
* In Equation 4, shouldn't the lower case be $x < 0$? Please check this.

---

> ### Author Response · Authors · 2026-03-03
> **Response to reviewer MExH (Concerns 1 and 2)**
>
> We thank you for a clear summary of our manuscript, and identifying many strengths including highlighting the novelty of the noise-preference phenomenon, the distinction between noise-in and noise-out networks, and our fixed-point shift-based analysis of zero-noise error. We also appreciate the positive feedback on the clarity of presentation, figures, and experimental analyses. We also thank you for your other constructive comments and hereby address these concerns.
>
> **Concern 1: Other training noise levels**
> > The results in Section 3.1 and Figure 2 seem to be for only one training noise level. I would expect to see a sweep over several different noise levels and a heat map characterising the summary of trends for such experiments. Currently, it is difficult to understand what would happen at very low or very high noise levels since only one training noise value is presented. In particular, I would have liked to see some argument relating this effect and overall performance to the effective signal-to-noise ratio (i.e., how the scale of the noise vis-a-vis the scale of the inputs or preactivations, etc. affects both task performance and noise level preference).”, "Please add a sweep over different training noise levels in addition to different testing noise levels."
>
> **Response:** We agree that some evaluation of the dependence of the noise preference phenomenon on the training noise level is important for understanding the phenomenon, and we had previously only considered one training noise level. We have performed additional experiments with the function computation task at various training noise levels; we have added a new figure panel containing the results, namely **figure 2b**. We have also refer you to the newly added discussion paragraph **“Noise as regularization”** (**section 4.1**).
>
> **Concern 2: Continuous attractor tasks **
> >"A key limitation, and this is acknowledged by the authors, is the focus on just tasks with fixed point dynamics. This is quite restrictive for a subset of tasks and there are several simplistic tasks that one could implement with RNNs, such as path integration tasks, which show more complex attractor dynamics (e.g., spatial navigation -> planar attractor, evidence integration -> line attractor, head direction -> ring attractor, etc.). It would be nice to see what happens in these tasks as well, and I would argue that these would be important to include for the paper to be complete + comprehensive.”, “Please add more tasks that do not just involve fixed point dynamics. Even simple continuous attractor tasks would in my opinion strengthen the paper (examples have been provided above).
>
> **Response:** Thank you for this comment. As you note, we had acknowledged that our initial suite of experiments, while generally broad, was still restricted to  fixed-point tasks. Though our new version includes additional broad range of tasks (like the multi-cognitive task suite, thanks to your comment), they are also fixed point tasks. For now, we have clearly stated that continuous attractor tasks are beyond the scope of the current paper. We agree, however, that such tasks, as well as other non-fixed-point tasks, constitute important future work, and we have edited the following text in Discussion **section 4.4** to reflect this:
> - _"We have shown that a noise preference emerges in noise-in networks trained on a few different tasks, but all of our tasks hinge on stabilizing network outputs around constant setpoints. As a result, our networks ended up relying on point attractors [refs] to implement their computations. However, many tasks require more than just stabilizing outputs around setpoints. One example is locomotion, where the outputs should exhibit the characteristics of a stable periodic motion or limit cycle [refs], not a point attractor. Another set of prominent examples are problems involving evidence accumulation or self-motion integration (i.e., path integration), which give rise to higher-dimensional continuous attractors [refs]. In these instances, given the relatively permissive conditions that we have outlined here, we hypothesize that noise preferences may emerge, but future work should examine a wider range of such tasks not easily solved by point attractor dynamics."_
>
> If you feel that this leaving discussion of continuous attractors for future work brings the paper below the threshold of acceptance, we are willing to implement an experiment with a simple continuous attractor task as you’ve suggested, before the full 4-week discussion period is over.

---

> > ### Comment · Reviewer_k7Bq · 2026-03-09
> >
> > Given TMLR's acceptance criteria, I would like to offer my opinion that limiting the analysis to fixed-point tasks is not grounds for rejection, as long as this limitation is acknowledged. The revised Discussion both acknowledges this limitation and makes predictions that can be tested in future work. This, in my view, is sufficient.

---

> ### Author Response · Authors · 2026-03-03
> **Response to reviewer MExH (Concern 3)**
>
> **Concern 3: Broader suite of cognitive tasks**
> >“For "neuroscientifically-relevant multi-task networks", i.e., the results in Section 3.2, I would have expected to see more task diversity rather than just one maze task with different start/end configurations being different tasks. The cited papers such as Yang et al. (2019) and Driscoll et al. (2024) show clear examples of RNNs jointly trained on a wide variety of tasks. It would be nice to see experiments with actual multitask RNNs such as this (e.g., on the ModCog suite – Khona et al. (2022) – https://github.com/mikailkhona/Mod_Cog).”, “If possible, multi-task networks with more task diversity and not just different configurations of the same task would be nice to include.”
>
> **Response:** Yes, we agree that our maze task doesn’t quite fulfill one’s expectations of a “multi-task network”. So we have implemented and performed additional experiments with six of the cognitive tasks from the suite used by  Yang et al. (2019) and Driscoll et al. (2024). The noise preference curves from these new multi-task experiments are in the newly added **figure 3c** . The methods for these tasks are in newly added **section 2.2.3** in the main manuscript and **appendix B.3**.  The results are elaborated in **section 3.2**.

---

> ### Author Response · Authors · 2026-03-03
> **Response to reviewer MExH (Concern 4)**
>
> **Concern 4: Noise-in versus noise-out networks**
> >“This is relatively minor among my weaknesses because I appreciate empirical work, but it would also be nice to have a theoretical understanding of why this phenomenon occurs, e.g., by trying to understand why noise compensation is learnt in noise-out networks (see Krishna et al. (2024), for example) but not for noise-in networks, causing them to overfit to and rely on the training noise level.”
>
> **Response:** We. agree that answering this key question is an important addition to this paper. We had provided some intuition for why this is so in our previous draft in the `single neuron regulator problem' (**section 3.5** now). We have expanded this here as follows:
>
> - **section 3.5** _"Since both noise-in and noise-out networks can exhibit noise-dependent fixed point shifts, but we only see a noise preference in noise-in networks, we hypothesized that there may be some performance incentive specific to noise-in networks that drives the development of a preference."_
>
> - **section 3.5** _"The single-neuron regulator experiments showed that there are regions in the parameter space where networks (both ReLU and tanh) exhibit a noise preference. When the pre-activation noise dominates the post-activation noise, the best networks tend to land inside or near these regions with a noise-preference, but when the post-activation noise dominates, they land squarely outside these regions (Figure 6a). We believe this trend is due to a tradeoff. If the stationary distribution straddles the saturation boundary, a portion of the pre-activation noise gets turned off, while leaving a portion of the post-activation noise out of reach for the recurrent weight to help in drawing the state back toward its expected value. For a network dominated by pre-activation noise, this is a good trade, so the stationary distribution should be placed near the boundary, but for a network dominated by post-activation noise, this is a bad trade, so the stationary distribution should be placed far from the boundary. This tracks with the observation that networks with higher magnitude recurrent weights tolerate higher inside noise levels before opting to place their setpoints near saturation boundaries. Given the reasoning in sections 3.3.1 and 3.3.2, this may explain why noise-in networks systematically develop a noise preference while noise-out networks do not. Further, this reasoning does not require that the activation function be rectifying, simply that it should be saturating, which is supported by the presence of the phenomenon in both ReLU and tanh single-neuron regulator experiments (Figure 6a-b)."_
>
> - **section 4.2** _The essential elements of our explanation are the following. 1) When network performance is quantified via error between outputs and deterministic targets, networks are incentivized to reduce the variance of their outputs, and by extension, any latents on which the outputs depend. 2) Saturating activation functions, $output=f(input)$, induce noise-level-dependent activation (output) biases for noisy input distributions if a significant fraction of the input probability mass lies beyond a saturation boundary. They do this by asymmetrically attenuating the influence of the input noise on the downstream output noise. Changing the variance of the input distribution shifts the output distribution, because increased probability mass in only one tail of the input distribution has an influence on the output distribution. This is in contrast to when the input distribution is in a linear region of the activation function domain, where both tails of the input distribution influence the output equally, so the mean of the output does not depend on the dispersion of the input. 3) When the input distribution straddles a saturation boundary, the output distribution has lower variance than when the input distribution is in a linear region of the activation function.
> Taken together, these key facts imply that saturating, noise-in networks trained to match their outputs to targets are incentivized to operate neurons near saturation as a way of reducing their output variance, and therefore, they are incentivized to operate in a way that makes the means of their output distributions dependent on their synaptic noise. Noise preference develops because networks learn to compensate for this noise-dependence in their outputs, but their compensation is only correct when the synaptic noise is similar to what they saw in training. This explanation does not apply to noise-out networks because the second and third key facts listed above do not apply to noise-out networks, but it does apply to many networks of interest to neural network research in general. We framed our exploration in the context of computational neuroscience, but this was not necessary._
>
> We slightly edited the abstract to better feature these discussions.

---

> ### Author Response · Authors · 2026-03-03
> **Response to reviewer MExH (Concerns 5-6)**
>
> **Concern 5: Neuroscientifically relevant noise-out networks**
> >“Another relatively minor point is that some neuroscientifically-relevant works such as Burak & Fiete (2012), Krishna et al. (2024), or Bredenberg et al. (2026) discuss noise-out networks unlike the others presented here. This could still be consistent with the rate-based interpretation of these models as long as the noise if Gaussian is smaller in scale than the firing rates themselves. In the first (Burak & Fiete, 2012), this view allows one to see memory drift as Brownian motion on a continuous attractor manifold. In the second (Krishna et al., 2024), noise is shown to be integral to the emergence of a neuroscientifically-relevant phenomenon (quiescent replay being a result of diffusion on the attractor manifold). The third (Bredenberg et al., 2026) shows denoising to be an important part of the computation and for improved quality of generated images. This explicitly relates to the point in the introduction that noise may have actual roles in computation itself (and not necessarily something to drown out), rather than being used in biological models solely to mimic synaptic noise or help alleviate training instabilities. It would be nice to add a discussion on this aspect and such works, especially given that a few of these works study tasks that would be relevant here (continuous attractor dynamics)”, “Please attempt to discuss the final point in my "weaknesses" about noise-out networks in computational neuroscience.”
>
> **Response:** Great point. We acknowledge that our initial draft framed noise-in formulations as essential for rate-based interpretations, when in fact, under the conditions you describe, noise-out formulations can work too. We have addressed this by editing the following text in **section 2.1**:
> - _"As presented thus far, the noise is injected inside the activation function, as is common in the computational neuroscience literature (e.g., Yang et al 2019, Driscoll et al 2024).  This placement ensures positivity of the neural state $h$, facilitating their interpretation as firing rates. However, some important modeling work in computational neuroscience uses noise injected outside the activation function (Buraka and Fiete 2012,Bredenberg et al 2026, Krishna et al 2024), as such formulations may still permit rate-based interpretation as long as neural activations are large in comparison to the noise. Thus, we also consider a variant of Equation 2 with the noise added outside the activation function ..."_
>
> On the other hand, though the noise was essential in reproducing/analysing phenomena observed in each of these studies, it did not necessarily become a vital part of the computations at test time, except in the case of Bredenberg et al. (2026), since this model had a “generative mode” that necessarily leveraged noise to sample novel instances from learned distributions. Nevertheless, we have edited our introduction as follows to include these works because we agree that they are relevant to our work to add important context:
> - _"One major reason for this is to model the noisy environment in which biological neurons must operate (refs), as multiple notable neural phenomena have been shown to result from mechanisms that only emerge to reject or otherwise address noise (e.g., Burak and Fiete 2012,Krishna et al, 2024}). Similarly, noise is essential for modeling the generative behavior of biological neural networks (Bredenberg et al 2026)."_
>
> **Concern 6: Typo in equation 4**
> > “In Equation 4, shouldn't the lower case be x<0? Please check this.”
>
> **Response:** Good catch, this was indeed a typo, and we’ve fixed it in the current revision. The upper case is <= 0 and the lower case is >0.

---

> > ### Comment · Reviewer_MExH · 2026-03-10
> >
> > I thank the authors for their detailed response. Here are my comments:
> > * Thank you for adding Figure 2b with a wider range of training noise values. Could you also comment on what would happen when sweeping over noise values for noise-out networks?
> > * I do not think the lack of non-fixed point tasks as grounds for outright rejection and I appreciate the explicit acknowledgment of this limitation. However, I do think the generality of the paper's claims can be shown even with a simple continuous attractor task (which I think is not too hard to implement). The paper title ("Paradoxical noise preference in RNNs") and claims (Intro: "Contrary to this expectation ... conditions under which a preference naturally develops") are quite general and not scoped to fixed point tasks alone, while the analysis relies only on such tasks. The acceptance criteria state that gaps between claims and evidence can be addressed through experiments or by adjusting (e.g., reducing [the scope of]) the claims. Given the ease of implementing a simplistic continuous attractor task, I think it should be possible for the authors to uphold the generality of their claims (as not just specific to fixed point task RNNs but potentially RNNs solving different kinds of tasks) + relevance to computational neuroscience (where such continuous attractor tasks are quite common) by running this experiment.
> > * Thank you for the experiment with additional cognitive tasks, this is encouraging to see.
> > * Thank you for the updated discussion, clarifications, and typo fixes.
> >
> > Overall, I am inclined to accept the paper (and have been since my initial review) but would appreciate if the authors could address point 2in particular as I believe it would be important to validate the generality of the claims here.

---

> > > ### Comment · Reviewer_MExH · 2026-03-15
> > >
> > > ### **Follow-up comment**
> > > Following up on my earlier comments, I attempted to reproduce some results from the paper and try out simple continuous attractor tasks. The supplementary material did not contain code and only had LaTeX sources (I would advise the authors to check this), so I implemented a simple CTRNN training setup myself. Here are my points (and I apologise that they are quite late during the discussion phase):
> > > * I was able to reproduce the noise preference results in fixed point networks (I tried $x = \sin(x)$ alone for a quick reproducibility check) -- in doing so I realised that the authors' figures and the paper in general does not show results over multiple seeds/initialisations/runs. This would be important to show that the results are not a one-off and not highly sensitive to random initialisations or seeds. In my runs I did in fact find the results in fixed point tasks to not be highly variant across seeds, so I would just request the authors to consider updating their figures to tighten up the experimental rigour here.
> > > * I also implemented simple evidence integration (line attractor) and head direction integration (ring attractor) tasks which are common in computational neuroscience. In these small experiments, I found that the noise preference phenomenon was less drastic with continuous attractor tasks compared to fixed point tasks, and also required higher training noise levels to become apparent for certain attractors (in particular the ring attractor). Furthermore, the preferred noise seemed lower in several runs than the actual training noise level. There could be other factors at play and my experiments were not very comprehensive -- but it is possible that the geometry of the attractors has some relationship to their robustness even when noise is removed in the noise-in networks.
> > > * Some encouraging signs include: noise-out networks showed a mostly monotonic increase in error as the testing noise increased from 0, irrespective of what the training noise level was; even in noise-in networks without a clear noise preference, the test error remained fairly flat from test noise $\sigma$ = 0 to a certain value (a bit lower than the training noise $\sigma$) and increased beyond that point.
> > >
> > > While I'm not sure I can share images/links/etc. on OpenReview, I've attempted to summarise my findings below if they seem helpful to the authors. I hope the authors can report their own results and respond to my comments.
> > >
> > > ### **Summary of experiments**
> > >
> > > **Network architecture:** Similar CTRNN architecture (ReLU, $\gamma = 0.2$, 128 neurons, linear readout).
> > >
> > > **Tasks:**
> > >
> > > * Sine computation (fixed point) -- attempt to reproduce Figure 2a from the paper
> > > * Evidence integration (line attractor) -- integrate noisy scalar evidence, target = $\tanh(\sum x_t)$
> > > * Head direction (ring attractor) -- integrate angular velocity, target = $(\cos\theta, \sin\theta)$
> > >
> > > **Setup:** For each task, I trained noise-in and noise-out networks at training noise levels $\sigma_{\text{train}} \in \{0, 0.1, 0.2, 0.3, 0.5\}$, and then computed the RMSE across test noise levels $\sigma_{\text{test}} \in [0, 0.6]$. The evidence integration task initialised the hidden state to zeros, while the head direction task used the encoder to map the initial angle cue directly to the hidden state. All other hyperparameters matched the paper to the best of my knowledge (Adam, MSE loss, learning rate schedule). Three seeds were used for the key comparison at $\sigma_{\text{train}} = 0.3$.
> > >
> > > **Results:** Noise preference emerges in all three attractor types for noise-in networks, but not for noise-out:
> > >
> > > | Task | $\sigma_{\text{train}}$ | RMSE($\sigma_{\text{test}}$=0) | Min RMSE | Best $\sigma_{\text{test}}$ | Improvement |
> > > |------|:-:|:-:|:-:|:-:|:-:|
> > > | **Sine (fixed point)** | 0.1 | 0.178 | 0.077 | 0.10 | 57% |
> > > | | 0.3 | 0.612 | 0.102 | 0.30 | 83% |
> > > | **Evidence integration (line)** | 0.1 | 0.125 | 0.092 | 0.08 | 26% |
> > > | | 0.3 | 0.148 | 0.123 | 0.14 | 17% |
> > > | **Head direction (ring)** | 0.3 | 0.163 | 0.157 | 0.10 | 4% |
> > > | | 0.5 | 0.268 | 0.218 | 0.36 | 19% |
> > >
> > > For continuous attractor tasks, these were mildly U-shaped RMSE curves with minimum near but slightly lower than $\sigma_{\text{train}}$. Noise-out networks showed no preference in any task (RMSE monotonically increasing with test noise), consistent with the paper's findings.

---

> ### Author Response · Authors · 2026-03-15
> **Initial response to noise preference in continuous attractors (reviewer MExH)**
>
> Thank you for your kind note. To address your earlier comments, we had also implemented a network with a continuous attractor for a 2D path integration task (non-fixed-point task), and will be uploading a revised version of the manuscript with these results included in the next couple of hours, aside from addressing other reviewer comments. We did find a noise preference here as well.
>
> Regarding your note (in these most recent comments) about the effect of the results on different noise realizations/network initializations, depending on which version you mean, this is already reflected in our extensive noise-preference plots from our previous versions. The experimental noise-preference plots of Fig 2 and Fig 3 are all results over multiple noise realizations: in every sub-plot, we indicate (1) the RMS error over multiple noise realizations (2) the mean error over multiple noise realizations and (3) the error standard deviation across multiple noise realizations. The error standard deviation provides the information you seek, quantifying the trial to trial variability due to completely different noise realizations for a given trained network. We will make sure to draw attention to this in the revision we soon post by adding a few lines.  If you are referring to different training runs, yes, in our experience, the results were across different training runs and their initial seeds/noise realizations. We will add an experiment/figure to show this. Thanks for the suggestion.

---

> > ### Comment · Reviewer_MExH · 2026-03-15
> >
> > >  If you are referring to different training runs, yes, in our experience, the results were across different training runs and their initial seeds/noise realizations. We will add an experiment/figure to show this. Thanks for the suggestion.
> >
> > Yes, I am specifically referring to different training runs. I understood that your existing figures quantified variability over 'trials' or samples + noise realisations but they do not currently show that the phenomenon is consistent over different training runs altogether, which would be important to demonstrate that the phenomenon is not sensitive to specific training initialisations (because a 'lucky' configuration was found, etc.).
> >
> > > To address your earlier comments, we had also implemented a network with a continuous attractor for a 2D path integration task (non-fixed-point task)...
> >
> > Thank you, I look forward to these results.

---

> ### Author Response · Authors · 2026-03-15
> **Second response to reviewer MExH's comments**
>
> **Comment 1: multiple training noise levels for noise-out networks**
> We have now added Figure A2 in the appendix that shows that there is no non-zero noise preference across training noise levels for noise-out networks. We have referenced this figure from the main manuscript as well.
>
> **Comment 2: Continuous attractor tasks**
> We have completed additional experiments with a continuous attractor task: the 2D path integration task of Cueva & Wei (2018). As we speculated in the previous revision, we observed the same noise preference phenomenon as in our other networks. The results of the new experiment may be found in Figure 3d (Figure 3d was formerly for the non-RNN feedforward network experiment, which is now Figure 3e). This new experiment is covered in the methods (Section 2.2.4, briefly explaining the task), the results (adding/adjusting a few lines in Section 3.2 to state the findings), discussion (Section 4 to include the experiment in the summary of findings, 4.2 to claim the new results imply generality, 4.4 to adjust our future work to no longer include continuous attractor style tasks and reframe the future work tasks as focussing on attractors with non-neutral stability on the manifold), and the supplemental appendices (both A to cover training details and B.4 to cover task details). Thank you for your insistence on including such a task. We agree that it suggests a greater degree of generality and makes the paper more relevant to the neuroscience community.
>
> **Comments 3-4:**
> We are glad you appreciated the additional experiments with the multi-cognitive suite, and thank you for the suggestion.
>
> **Comment: Effect of multiple noise realizations / different training runs**
> This is regarding your comment from earlier today!
>
> Our results in Figure 2, Figure 3, etc. quantify the effect of multiple noise realizations by showing how the error standard deviation due to multiple noise realizations changes with test noise and training noise. We have now added an explicit note to this effect in the Figure 2 captions. Of course, these are different noise realizations for a fixed converged network (although you have now clarified that this you meant the issue in the next paragraph, it seemed worth emphasizing this anyway).
>
> Regarding networks from different training runs from different random network initializations/noise realizations, we have found that the noise-preference effect is qualitatively quite robust to such initializations and training runs. To support this, we have now produced a
> new appendix figure A1 to show the small variance in the noise preference curves for networks obtained from different training runs.

---

> > ### Author Response · Authors · 2026-03-15
> > **Comment: Code in Supplementary Material**
> >
> > We have now also ensured that the code is uploaded as a zip file as Supplementary Material. Thank you.

---

> > > ### Comment · Reviewer_MExH · 2026-03-16
> > >
> > > Thank you very much for including these additional figures and results. I think they improve the generality of the paper's claims and the overall experimental thoroughness/rigour; I appreciate the authors for considering my suggestions and acting on them. My reproduction of these results, now including the path integration task as well, align with the overall results presented.
> > >
> > > A small note would be that in Section 3.2, you mention that the preferred noise is close to but 'further away' from the training noise level. It seems that the preferred noise is typically lower than the training noise level in these experiments and it would be better to state this unambiguously if that is the case.
> > >
> > > I also appreciate the authors for uploading the code. I am not a Matlab expert but the implementation details seemed to align with my reproduction and to my knowledge, seemed to be correct.
> > >
> > > Thank you again for your efforts. As I stated earlier, I remain inclined to accept this paper, now more confidently.

---

> > > > ### Author Response · Authors · 2026-03-17
> > > > **Sentence added in Section 3.2**
> > > >
> > > > Thank you for the kind words and appreciation of our edits. To address your small note above, we edited the sentence in section 3.2 to: _"although, in detail, the optimal performance for the multi-cognitive task suite and the path integration was further away from and generally smaller than the training noise level."_
> > > >
> > > > and added the following sentence as a further remark: _"Specifically, in these RNN experiments, we find that the error bias is optimal at nearly the training noise level and the error standard deviation is monotonic increasing: these two observations imply that the total error will have a minimum to the left of the training noise level as seen in these experiments."_
> > > >
> > > > and the following remark about our feedforward network: _"Unlike in the RNNs, for these feedforward networks, the error bias was minimized at slightly higher than the training noise level."_
> > > >
> > > > We have uploaded a new manuscript version with these minor edits.

---

> > > > > ### Comment · Reviewer_MExH · 2026-03-18
> > > > >
> > > > > Thanks again!

---

### Author Response · Authors · 2026-03-02
**Update on an upcoming revision later today**

We thank the reviewers for their careful, constructive, and largely positive feedback and assessments! All three reviewers highlighted several common strengths of the submission, including the novelty of the noise-preference phenomenon, the clarity of presentation and figures, and the converging empirical evidence across multiple tasks, while also offering a number of thoughtful suggestions that we believe will substantially strengthen the paper. We are preparing a comprehensive revision addressing the requested additions (e.g., broader task coverage via new numerical experiments, additional analyses, clarifications, and expanded discussion) and will post it by sometime March 2 (i.e., later today). We would appreciate keeping the discussion period open for the full four weeks (ie until March 17) so that we have sufficient time to incorporate any follow-up comments from the reviewers and respond to them carefully and thoroughly; we will actively monitor the discussion and continue engaging with reviewers throughout this period.

---

> ### Author Response · Authors · 2026-03-03
> **Revised manuscript posted (overall response)**
>
> Dear reviewers - We have now posted a revision to the manuscript addressing the reviewer comments. Our responses are posted as separate comments underneath your respective reviews. We look forward to any further notes/feedback from the reviewers, and in case there are any, we would appreciate keeping the discussion period open for the full four weeks (ie until March 17) so that we have sufficient time to incorporate any such follow-up comments from the reviewers in a careful and thoughtful manner. We will actively monitor the discussion and continue engaging with reviewers throughout this period.

---

### Author Response · Authors · 2026-03-15
**Upcoming revision with all further reviewer comments addressed**

Dear all - We thank the action editor for the guidance and overview, and we thank the reviewers for the second round of comments, all of which we found to be helpful and reasonable, as in the first round. In the next couple of hours or so, we will post a revised manuscript that comprehensively addresses all the reviewer comments. We believe this review process has resulted in a much stronger paper of broader interest, so we appreciate all the reviewers. We have added comments under each reviewer's second round of comments, addressing them point-by-point. This includes:
- Added new experiments (figure and additional writing) showing that a continuous attractor network also has noise preference, as requested by reviewer MExH
- Added new experiments (figure) showing that noise-out networks do not seem to develop non-zero noise preference across a range of training noise magnitudes by MExH
- mirroring the section in the Discussion, expanded heuristic explanation of why noise-in networks develop noise-preference but not noise-out networks, even though both have fixed point shifts due to noise as requested by reviewer Y2Tt
- other minor edits addressing reviewer comments

Thank you again!

---

> ### Author Response · Authors · 2026-03-15
> **Second revision posted with all reviewer comments addressed**
>
> We thank the action editor for the guidance and overview, and we thank the reviewers for the second round of comments, all of which we found to be helpful and reasonable, as in the first round. We have posted a revised manuscript that comprehensively addresses all the reviewer comments, including reviewer comments from earlier today. We believe this review process has resulted in a much stronger paper of broader interest, so we appreciate all the reviewers. We have added comments under each reviewer's second round of comments, addressing them point-by-point.
>
> In brief, changes include:
> - Added new experiments (figure and additional writing) showing that a continuous attractor network also has nonzero noise preference.
> - Added new experiments and figure showing that noise-out networks do not seem to develop non-zero noise preference across a range of training noise magnitudes.
> - Added new experiments and figure showing that the noise-preference behavior/statistics are qualitatively the same and quantitatively similar across networks from different training runs (addressing new reviewer comment from earlier today)
> - Mirroring the section in the Discussion, expanded heuristic explanation of why noise-in networks develop noise-preference but not noise-out networks, even though both have fixed point shifts due to noise
> - Other minor edits addressing reviewer comments
>
> Thanks again for a remarkably constructive and fruitful review process and discussions!

---

### Decision · Action_Editor_7Lkp · 2026-03-20

**Recommendation:** Accept as is

**Audience:**

Yes

**Audience Explanation:**

The reviewers agreed that the paper will be of interest, especially to the computational neuroscience community.

**Claims And Evidence:**

Yes

**Claims Explanation:**

All reviewers agreed that the paper presents clear and solid evidence, and recommended publication. The authors have addressed the main concerns raised during the review period.

While the submission does not provide a full theoretical explanation for the differences between noise-out and noise-in settings, the findings are convincingly supported by numerical experiments.

---

> ### Author Response · Authors · 2026-05-03
> **Camera ready version posted**
>
> Dear Action Editor - Thank you for the decision, editorial overview, and guidance. We have now posted a camera ready version of the manuscript. We have made no changes except to make the manuscript camera ready and link to a github repository of the code (while the code is also available as a Supplementary File with the submission). Thank you again!